# Non-polar ether-based electrolyte solutions for stable high-voltage non-aqueous lithium metal batteries

Zheng Li [1] ✉, Harsha Rao[1], Rasha Atwi[2], Bhuvaneswari M. Sivakumar[3], Bharat Gwalani [3,4], Scott Gray[5], Kee Sung Han [3,4], Thomas A. Everett [6], Tanvi A. Ajantiwalay[3], Vijayakumar Murugesan [3,4], Nav Nidhi Rajput[2] & Vilas G. Pol [1] ✉

The electrochemical instability of ether-based electrolyte solutions hinders their practical applications in high-voltage Li metal batteries. To circumvent this issue, here, we propose a dilution strategy to lose the Li$^+$/solvent interaction and use the dilute non-aqueous electrolyte solution in high-voltage lithium metal batteries. We demonstrate that in a non-polar dipropyl ether (DPE)-based electrolyte solution with lithium bis(fluorosulfonyl) imide salt, the decomposition order of solvated species can be adjusted to promote the Li$^+$/salt-derived anion clusters decomposition over free ether solvent molecules. This selective mechanism favors the formation of a robust cathode electrolyte interphase (CEI) and a solvent-deficient electric double-layer structure at the positive electrode interface. When the DPE-based electrolyte is tested in combination with a Li metal negative electrode (50 μm thick) and a LiNi$_{0.8}$Co$_{0.1}$Mn$_{0.1}$O$_2$-based positive electrode (3.3 mAh/cm$^2$) in pouch cell configuration at 25 °C, a specific discharge capacity retention of about 74% after 150 cycles (0.33 and 1 mA/cm$^2$ charge and discharge, respectively) is obtained.

Pairing Li metal anode (LMA) with high voltage Ni-rich Li(Ni$_x$Co$_y$Mn$_{1-x-y}$)O$_2$ (NCM) cathodes for next-generation Li metal batteries (LMBs) is one of the most promising approaches for increasing the energy density of the battery to meet the ever-increasing demand for electric energy storage[1–3]. However, LMA suffers from problematic cyclability due to its volume change and dendritic deposition[4]. The continuous cracking and regeneration of the solid electrolyte interphase (SEI) favor the lithium inventory loss at the Li metal electrode and non-aqueous electrolyte, leading to "Dead Li" (i.e., Li metal regions which are electronically disconnected from the current collector) formation, which accelerates capacity loss and mechanical degradation[5]. Meanwhile, electrolyte oxidation and other parasitic

reactions at the cathode electrolyte interphase (CEI) of Ni-rich NCM cathodes also promotes battery failure[6]. Tailoring the electrolyte solution properties is generally an efficient pathway for regulating LMBs' interfacial behaviors, mitigating side reactions to enable their long-term operation[3].

Ether solvents like 1,2-Dimethoxyethane (DME) give rise to better LMA compatibility (Coulombic efficiency (CE)) than conventional carbonates[7]. However, their oxidation instability prevents high-voltage battery applications above 4 V[8]. Besides, dilute ether electrolytes (<3 M)[9] with imide-type salts such as lithium bis(fluorosulfonyl) imide (LiFSI) or lithium bis(trifluoromethanesulfonic)imide (LiTFSI) corrode Al current collector above 3.8 V and deteriorate battery performance[10].

[1]Davidson School of Chemical Engineering, Purdue University, West Lafayette, IN 47907, USA. [2]Department of Materials Science and Chemical Engineering, Stony Brook University, Stony Brook, NY 11794, USA. [3]Pacific Northwest National Laboratory, Richland, WA 99352, USA. [4]The Joint Center for Energy Storage Research, Pacific Northwest National Laboratory, Richland, WA 99352, USA. [5]Battery Innovation Center, Newberry, IN 47449, USA. [6]Discovery Park, Purdue University, West Lafayette, IN 47907, USA. ✉e-mail: li3259@purdue.edu; vpol@purdue.edu

Several strategies have been reported to improve their oxidation stability. Highly concentrated (>4 M) ether electrolytes (HCE) containing minimal free solvent molecules extended their anodic potential window[2]. Anion-involved solvation structure adjusts the CEI chemistry, generating an inorganic-rich passivation layer, which prevents further decomposition[8,11]. Incorporating noncoordinating hydrofluoroethers (HFEs) into HCE as diluents can preserve the Li$^+$ solvation environment while simultaneously dividing large ion aggregates into small clusters to reduce the viscosity, yielding the concept of locally high concentrated electrolyte (LHCE)[12]. Besides, molecular engineering via partially fluorinating ether molecules can improve ethers' anodic stability and tune the SEI/CEI composition for improved stability[13,14].

However, the abovementioned pathways inevitably increase the manufacturing cost of the electrolyte. The high density of HFEs (>1.4 g/ml at 25 °C)[15] also has an adverse impact on energy density[16]. The uncertain environmental effects of fluorinated ethers could be potential obstacles to large-scale commercialization[17]. Therefore, designing optimal electrolytes to overcome anodic stability issues of ethers via facile and cost-effective approaches should still be considered, despite the rarely reported stable high-voltage LMB with dilute nonfluorinated ether[18]. Moreover, the prerequisites for building stable ether-based electrolytes are still unclear. The improved performance has been attributed to cathode passivation, molecular stability, Al corrosion prevention, and solvation structure, but the correlations among these crucial factors have been seldom elucidated. The fundamental oxidation behavior of diluted ether-based electrolyte solutions on a molecular level and their interfacial evolutions at high-voltage cathodes are also rarely interpreted. These ambiguities hinder the precise design of ether electrolyte systems, especially with low salt concentration (<3 M)[3,9].

Recent works underline investigating the solvation behavior of electrolytes and its correlation with battery performance[19]. Specifically, several works reported weakly-solvated ether electrolytes (WSEE) featuring an anion-involved solvation environment with 1.0 M salt concentration[20,21]. Less polar ethers compared to conventional glyme-based ethers allow suppressed Li$^+$-solvent interactions and lead to the formation of contact ion pairs (CIPs) in the solution[21,22]. Some improved oxidation stabilities have been identified with dilute WSEEs. For example, Holoubek et al. observed better stability of 1 M LiFSI in diethyl ether (DEE) on the Al electrode and attributed it to CIP structures[23]. Chen et al. and Pham et al. separately reported improved performance of 1,2-diethoxyethanes over DME with more than 3.8 M LiFSI concentration, due to the anion-enriched solvation and the derived better CEI[24,25]. However, all studied WSEE systems still failed to deliver satisfactory NCM cathode compatibility with low salt concentration.

Here we systematically explore the high voltage compatibility of dilute LiFSI-based, non-aqueous electrolytes in nonfluorinated ether solvents. A series of ether solvents with decreasing coordinating power including diglyme (DIG), DME, DEE, and dipropyl ether (DPE), were investigated. We demonstrated that low concentration[9] (1.8 M LiFSI) ether-based electrolyte successfully endures long-term high voltage (4.3 V) operation of practical LMB (with controlled negative/positive (N/P) ratio and lean electrolyte condition) when using the highly nonpolar DPE as solvent. We also confirmed essential correlations between dilute ether-based electrolytes' solvation behavior and their stability on high-voltage LiNi$_{0.8}$Co$_{0.1}$Mn$_{0.1}$O$_2$ (NCM811)-based positive electrode, including oxidation pathway, passivation behavior, and Al current collector corrosion. The correlations were further interpreted via detailed classical molecular dynamics (MD) simulations and density functional theory (DFT) calculations coupled with multimodal experimental analyses. It was demonstrated that improving the compatibility of ether electrolytes with high voltage cathode does not necessarily require thermodynamically improved oxidation stability via conventional approaches, such as diminishing uncoordinated ether

molecules or introducing molecular fluorination. Rearranging the degradation order of solvation species in the electrolyte and adjusting the composition of the electric double layer on the positive electrode surface can kinetically stabilize the electrode|electrolyte interface.

## Results

A series of standard non-fluorinated ether solvents including DPE, DEE, DME, and DIG were selected as representatives of mono-, bi-, and tridentate ethers, possessing increasing chelating effect to Li$^+$ in the electrolyte (Fig. 1a, b, Supplementary Figs. 1, 2, Table 1, and Note 1). All the tests throughout this work were performed at 25 °C and all the electrolytes were prepared with LiFSI salt, unless specified. The monodentate ethers (DPE and DEE) with smaller molecular size exhibit consistently smaller density, surface tension and viscosity (compared to their multidentate analogs), which favors the wetting ability on the porous positive electrode and separator surface[26]. Compared to DEE, the DPE molecule has longer alkyl groups and thus weaker coordinating ability due to stronger steric hindrance[24]. Therefore, the four ethers with the solvating power order of DPE < DEE < DME < DIG were investigated regarding their high-voltage compatibility. In Fig. 1c, d, four electrolytes with 1.8 M LiFSI salt were subjected to preliminarily anodic stability tests, where improved high-voltage performance was demonstrated with low polarity ethers. Ohmic drop (IR) Corrected Linear scanning voltammetry (LSV) was conducted with Li||Al coin cells. Figure 1c and Supplementary Fig. 3 show that both DEE and DPE exhibit stability up to 5.8 V with minimal anodic current generation, while DME and DIG electrolytes display significant oxidation reactions from 4.5 and 4.1 V, respectively (Supplementary Note 2). More importantly, applying the four electrolytes to Li||NCM811 (1.6 mAh/cm$^2$) coin cells generated stark differences in the cathode CE (Fig. 1d). The 1.8 M DPE electrolyte shows an high average efficiency of 99.92%, which has been seldom reported with dilute ether electrolytes in nonfluorinated solvent[18]. DEE also displays an improved CE of 99.58% compared to 1.8 M DME (98.54%) and DIG-based (98.72%) electrolytes. Chronoamperometry study of Li||NCM811 coin cells at 4.3 V (Fig. 1e) identified the lowest anodic leakage current (2.1 μA) from DPE electrolyte, with the order of DPE < DEE < DIG < DME. The high-voltage compatibility of the DPE-based electrolyte is also comparable to the reported state-of-the-art electrolytes[27], which is even higher than 4.8 M concentrated DME electrolyte (Supplementary Fig. 4)[8]. The nonpolar DPE and DEE electrolyte enables stable cycling at 1.6 mA/cm$^2$ (Fig. 1f and Supplementary Fig. 5), while the DME and DIG cell experienced fast capacity fading and the sign of early battery failure at 3.2 mA/cm$^2$. Overall, the results suggest strong correlations between electrolyte stability and solvent solvating power. Therefore, their solvation structures need to be identified in priority to help in interpreting the different performance.

### Solvation structures and the correlations to solvent coordination ability

The solvation structures of the 1.8 M LiFSI ether electrolytes with decreasing solvent solvation power were investigated via classical MD simulations. Radial distribution functions (RDF) of solvation structures are shown in Fig. 2a–c. The Li$^+$-solvent and Li$^+$-FSI$^-$ interactions exhibit opposite trends in each electrolyte system. Polar DME and DIG solvents strongly coordinate with the cation, as evident from the sharp RDF peaks in these systems. In contrast, the negligible Li$^+$-FSI$^-$ RDF peaks indicate that the anion does not contribute significantly to the Li$^+$ solvation structure, especially in the DIG-containing electrolyte[28,29]. On the other hand, nonpolar DPE and DEE ethers exhibit weak solvating power that enables Li$^+$ to strongly interact with the anion in the primary solvation shell ($r \approx 1.8$ Å)[23]. It is worth noting that the strength of the Li$^+$-solvent interactions increases monotonically with the solvent dielectric constant. Additionally, shorter interaction distances between Li$^+$ ions in WSEE systems (Fig. 2c) suggest the formation of

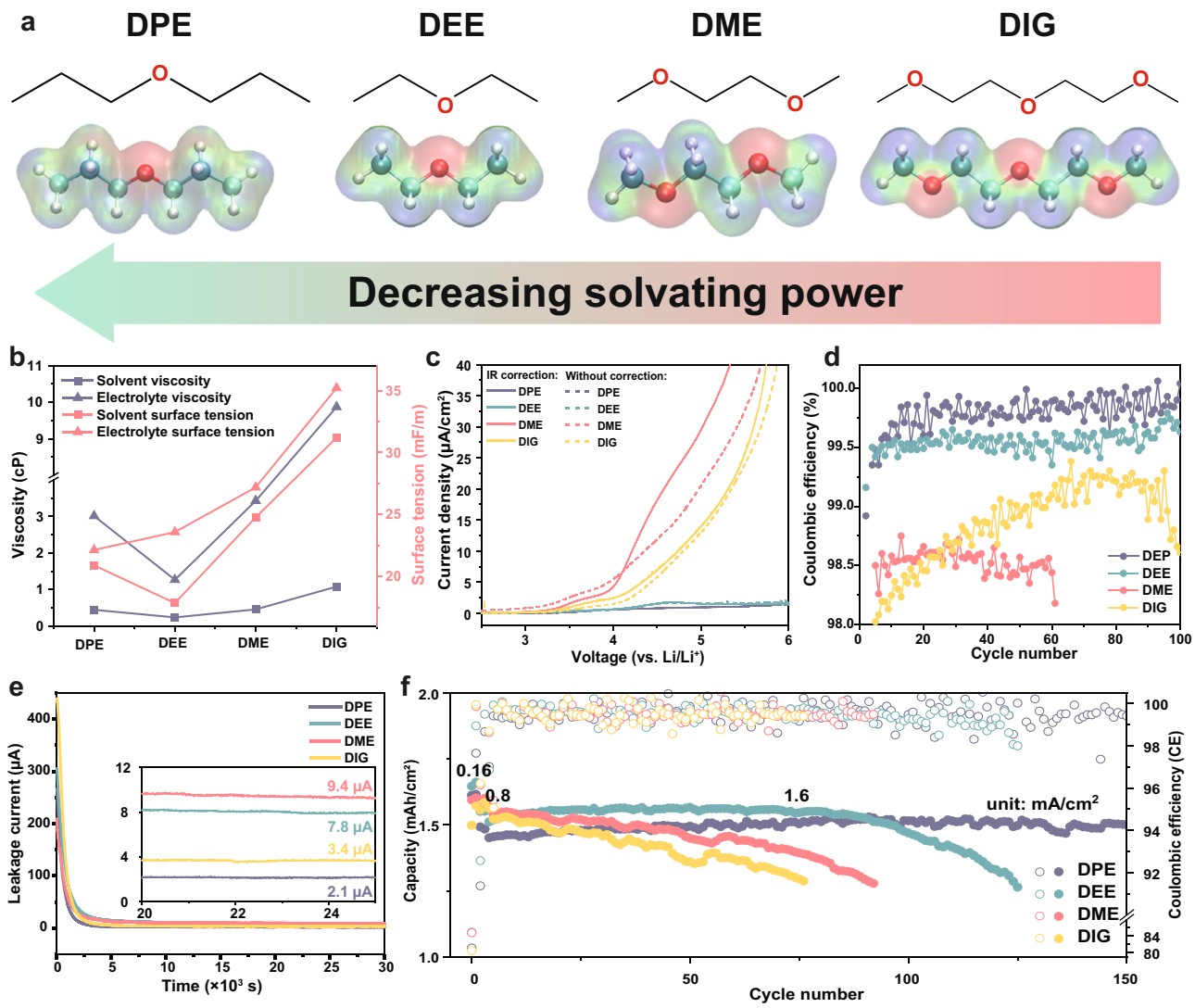

**Fig. 1 | Ether-based electrolytes with decreasing solvating power and their anodic stability.** All the electrolytes contain 1.8 M LiFSI, all the tests were performed at 25 °C. **a** The electrostatic potential (ESP) maps of the studied ether molecules. The red, green, and white spheres stand for the O, C, and H Atoms, respectively. **b** The viscosity and surface tension values of the studied electrolytes and solvents. **c** LSV results of Li‖Al coin cells at the scan rate of 0.5 mV/s. **d** Coulombic efficiencies of Li‖NCM811 coin cells cycled between 2.8 and 4.3 V at 0.48 mA/cm². Cathode loading is around 8.21 mg/cm². **e** Chronoamperometry testing of Li‖NCM811 coin cells at 4.3 V. **f** Long-term cycling performance of Li‖ NCM811 coin cells at 1.6 mAh/cm². Two formation cycles at 0.16 and 0.8 mA/cm² were performed.

large ion aggregates (AGGs) consisting of multiple Li⁺ and FSI⁻ species[30]. The aggregate formation is driven by bridging coordination of FSI⁻ anions across multiple Li⁺ cations through the oxygen atoms. Such aggregation behavior is also confirmed by the long-distance Li⁺-FSI⁻ interactions within the secondary solvation shell ($r \approx 4.2$ Å), as shown in Fig. 2b. On the other hand, the cations are much more separated in polar ethers since they are tightly wrapped by solvent molecules (solvent separated ion pairs; SSIPs) due to their competitive coordination[20]. The snapshots of each simulation box in Fig. 2d also provide theoretical confirmation of this structural behavior. DME and DIG electrolytes feature homogeneously dispersed SSIPs, anions, and free ether molecules. In contrast, DEE shows a slightly localized accumulation of CIPs and AGGs, while Li⁺-FSI⁻ pairs strongly aggregate in DPE, the highest nonpolar ether among the studied solvents. Due to the pronounced AGGs, the conductivities of DPE and DEE electrolyte are also reduced (Supplementary Fig. 6). In the meantime, according to the results from Saito et al.[31], the strengthened microviscosity of the WSEE including their cation-anion and cation-polymer separator coulombic interactions slow down the ion diffusion in the electrolyte.

These reasons explain the suppressed ionic transport in the DPE electrolyte, its higher overpotential and slightly less capacity in the Li‖ NCM811 coin cells at each current densities (see Fig. 1f and Supplementary Fig. 5). The most probable solvation structures extracted from MD simulations (Fig. 2d and Supplementary Fig. 7) suggest that Li⁺ is most likely to be coordinated with four anions (36.6% occurrence) in the DPE electrolyte and three anions and one solvent molecule (46.4% occurrence) in the DEE electrolyte. Instead, the Li⁺ solvation shell in the DME-based electrolyte most likely (45.7% of occurrence) contains two solvent molecules and one anion at the studied concentration, while the Li⁺ is completely chelated by DIG molecules, with an average coordination number of six DIG oxygens.

Raman spectroscopy and nuclear magnetic resonance (NMR) measurements and analyses were carried out on each electrolyte system to confirm the solvation structures experimentally[32,33]. In Fig. 2e, Raman spectra of each ether with increasing salt concentration (molar ratio of salt to solvent from 1:9 to 1:1) were collected from the region containing FSI⁻ S–N–S (700–780 cm⁻¹) and ether C–O–C vibrations (800–950 cm⁻¹). The higher position of the S-N-S band indicates

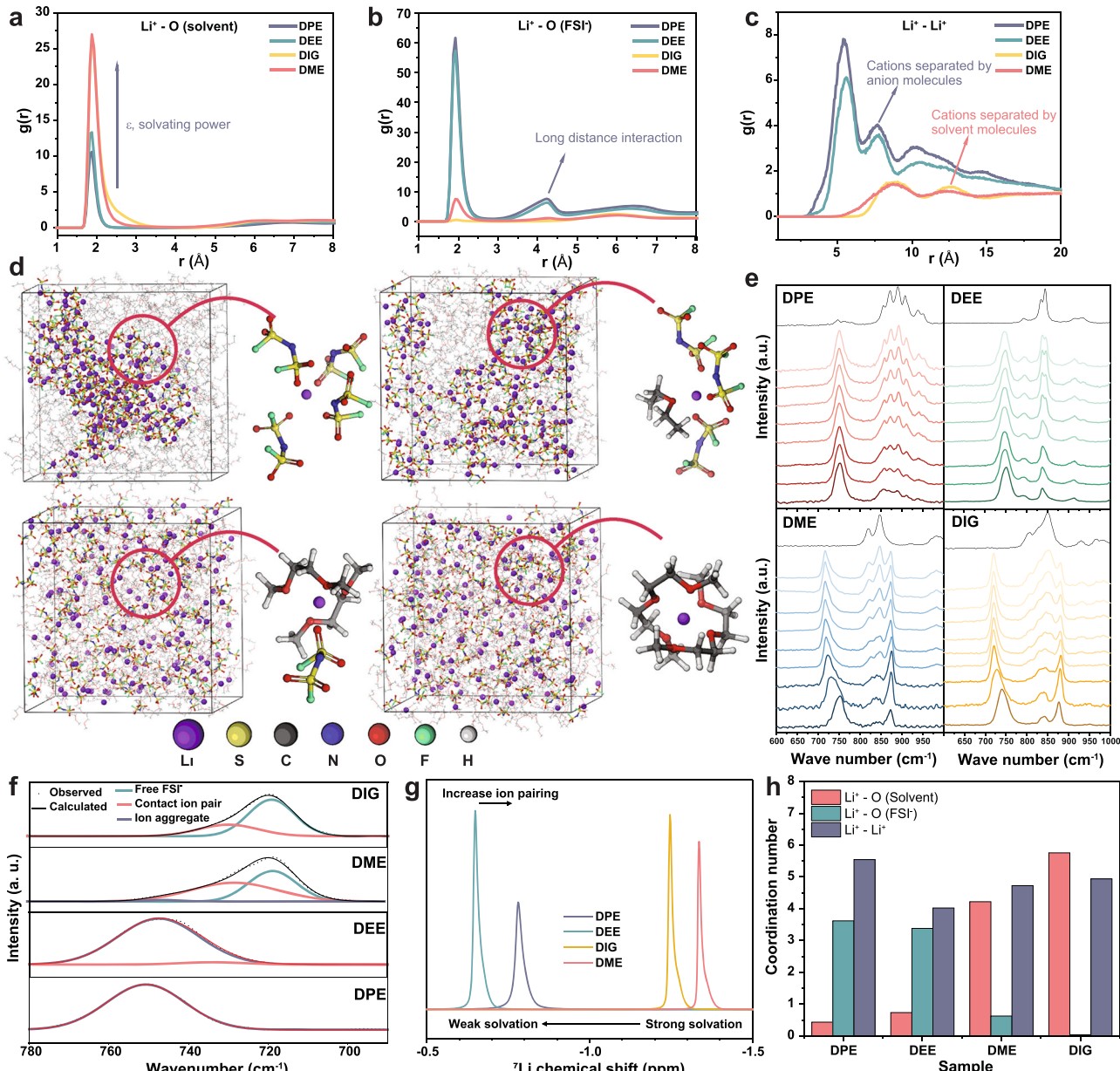

**Fig. 2 | Investigations on the solvation environment of the ether-based electrolyte solutions.** Radial distribution functions of interactions between Li$^+$ to solvent (**a**), Li$^+$ to FSI$^-$ (**b**), and Li$^+$ to Li$^+$ (**c**). **d** Snapshots of the MD simulation boxes of the four 1.8 M LiFSI electrolytes at 25 °C. **e** Concentration-dependent Raman spectra of electrolytes with salt to solvent ratio from 1:1 (1:2 for the DPE and DEE; bottom) to 1:9 (top). **f** Deconvolution analyses of S-N-S Raman spectroscopy signal from the 1.8 M electrolytes. $^7$Li NMR spectra (**g**) and atomic coordination numbers (**h**) of the four 1.8 M LiFSI electrolytes.

mitigated LiFSI salt dissociation[34]. Such signals from the DPE electrolytes ranged from 751 and 753 cm$^{-1}$, suggesting a minimum shift with increasing concentration, while DEE shows a slight shift from 746 to 752 cm$^{-1}$. However, significant band evolution occurs in DME (716–750 cm$^{-1}$) and DIG (720–740 cm$^{-1}$) electrolytes. Meanwhile, weaker solvating power also leads to abundant free solvent molecules regardless of concentrations. Free solvent signals were well detected in DEE and DPE electrolytes up to the highest salt concentration. Unlike DME and DIG systems, the two WSEEs exhibit a minor Li$^+$-ether interaction band at around 874 cm$^{-1}$. Specifically, deconvoluting results of S–N–S bands from 1.8 M electrolytes in Fig. 2f indicate a large dissociation degree of DME (83.4%) and DIG (86.1%) electrolytes while no free anions exist in DPE and DEE[20]. NMR results of the four electrolytes in Fig. 2g and Supplementary Fig. 8 also reveals the same tendency. The noticeable upfield (more negative) shift in $^7$Li signals from DME and DIG electrolytes compared to WSEEs (DPE and DEE) suggest strong

Li$^+$-ether coordination[24]. Stronger interaction with anions in DPE causes an upfield shift compared to DEE as similar with the upfield shift of DME compared with DIG, which also confirms its strengthened ion aggregation and strong Li$^+$–Li$^+$ and Li-anion interactions, as evident from the MD coordination numbers shown in Fig. 2h.

Overall, the experimental results are well-correlated with MD simulations. Highly nonpolar ethers such as DPE with largely reduced binding energy to Li$^+$ (Supplementary Fig. 9) can easily suppress salt dissociation and facilitate forming ion aggregates starting from 1 M concentration. The Li$^+$-anion coordination in dilute DPE is even more intense than in the highly concentrated DME electrolyte[24]. On the other hand, the fact that the anions undertake the coordination to Li$^+$ also implies the existence of more abundant free ether molecules which are predominantly regarded as the critical factor of low stability[35]. With these intriguing results, the following sections strive to interpret such improved high-voltage stability, especially considering the most stable

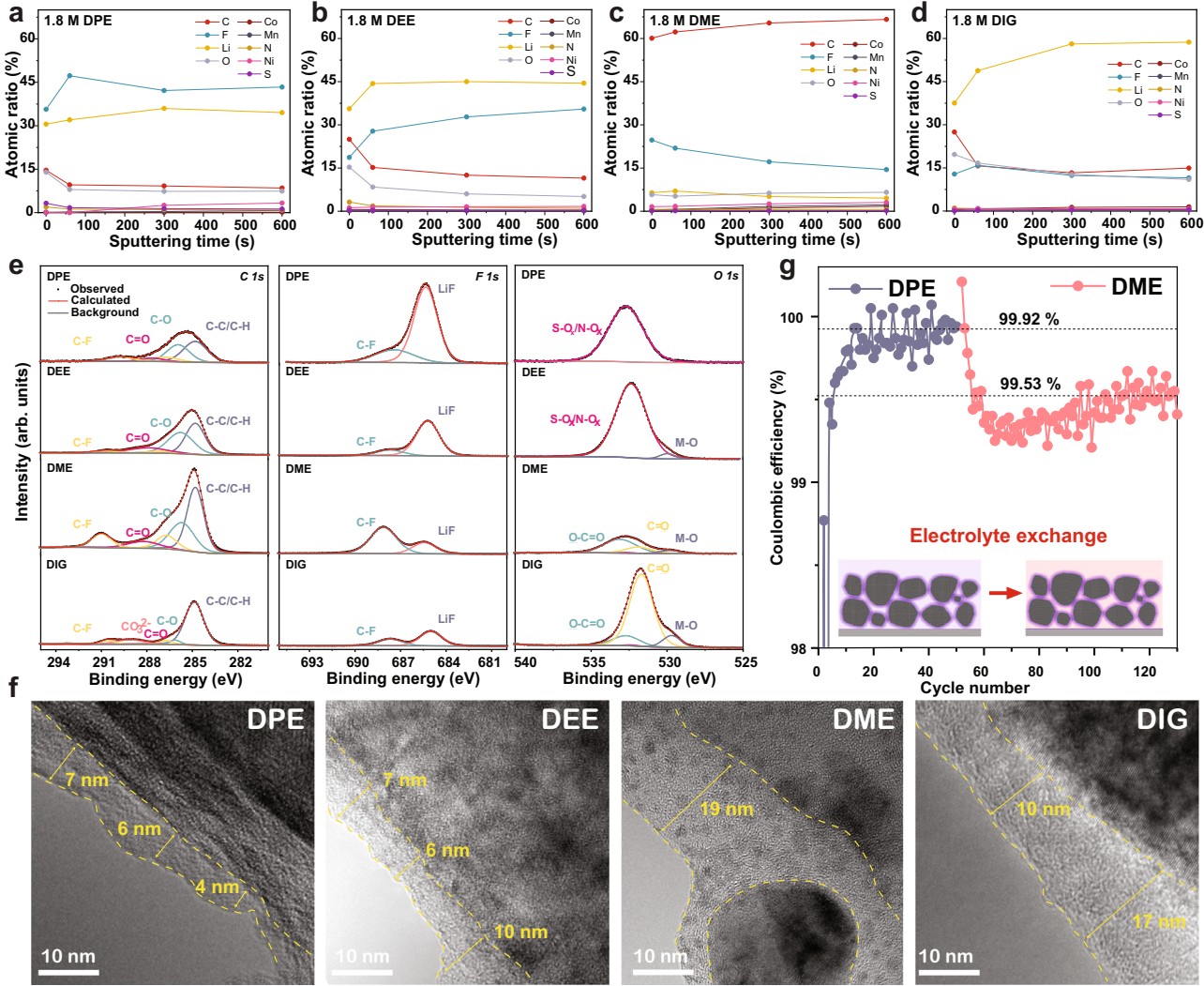

**Fig. 3 | Characterizations of the positive electrode CEI.** All the cells were cycled in the 1.8 M LiFSI electrolytes with different ether solvents at 0.48 mA/cm² and 25 °C. **a–d** XPS analyses of the depth-dependent elemental concentration on fully discharged NCM811 cathodes after 100. **e** XPS Fine spectra deconvoluting analyses of *C 1s*, *F 1s*, and *O 1s* signals. **f** High-resolution TEM imaging of CEI layers on cycled cathodes. The electrode samples were cut into two halves for the XPS and TEM. **g** Electrolyte exchange study between 1.8 M LiFSI in DPE and DME electrolytes within Li‖NCM811 coin cells. The schematic of the positive electrode with exchanged electrolytes is shown in the inset.

DPE electrolyte is the richest in free ether molecules. We attribute the improved performance to the following possible reasons, whose contributions and coupled interplay will be assessed in the discussions. (1) The CEI layer formed in WSEEs might be robust, which can prevent the parasitic reactions of the ether molecules on the cathode surface. (2) Monodentate ether molecules might be more stable than other ethers, which suppresses oxidation reactions. (3) The solvation behavior of the DPE electrolyte enables a different degradation pathway and interfacial behavior on the cathode surface to improve the stability.

## Cathode passivation and the influence on ether oxidation

The cathode passivation due to CEI formation is studied at first since it has been reported as the primary origin of the cathode stability[6]. An inorganic-rich CEI usually reduces cathode parasitic reactions such as electrolyte oxidation, cathode phase change, and transition metal dissolution[36]. Depth-dependent ex situ X-ray photoelectron spectroscopy (XPS) was employed to identify the interfacial chemical species of fully discharged NCM811 cathodes (1.6 mAh/cm²) after 100 cycles in the different electrolytes (Li‖NCM811 coin cells, 0.48 mA/cm²). Figure 3a–d shows the atomic concentration of each cathode, where

the DPE electrolyte displays an F-rich (~43%) passivation layer. In contrast, cathodes from DME and DIG electrolytes are covered by abundant carbon and oxygen species. Fine spectra deconvolution analyses in Fig. 3e suggest LiF is the dominant CEI compound from DPE electrolyte[37]. In contrast, the polar electrolytes generate an organic-rich surface layer with abundant C−O (286.15 eV) and C=O (287.8 eV) species. Additionally, only the DPE electrolyte exhibits complete cathode passivation, where the metal oxide signal (M−O, ~529 eV) is not observed[38]. S−$O_x$ was detected as the exclusive chemical species in the surface layer, which demonstrates the complete anion-derived CEI in the DPE sample. Depth-dependent profiles in Supplementary Fig. 10 show gradually increasing lattice oxygen concentration as Ar sputtering proceeds, which indicates that DPE formed a thin and uniform surface layer on NCM cathode (Supplementary Note 3)[35]. *Mn 2p* signals shown in Supplementary Fig. 11 are also only undetected in the DPE sample, with the intensity order of DPE < DEE < DIG < DME. These results confirm the improved cathode passivation with low polarity ethers. Prominent signals of *S 2p* in DPE and DEE cathode in Supplementary Fig. 12 also verify the anion decomposition. Abundant $SO_xF_y$ species at ~170 eV reflect the direct degradation of FSI⁻ anion on the cathode surface[30]. Ex situ high-resolution TEM was also used to

investigate the CEI formation on the same batch of electrode samples which were used for XPS (electrodes were cut into two halves for the two studies). In Fig. 3f, the DPE and DEE electrolytes exhibit thin and compact surface passivation layers with an average thickness of 5 and 8 nm, respectively. The DPE electrolyte especially shows crystalized inorganic species due to the anion decomposition, which is well-correlated with the XPS results. On the other hand, thick and amorphous layers were found on the DME (~19 nm) and DIG (~15 nm) cathode surface. It demonstrates these electrolytes cannot passivate the NCM cathode with the abundant organic products to prevent the continuous parasitic reactions. Moreover, isolated small NCM particles are also observed in the DME sample, which verifies the poor protection from its CEI and the resulted positive electrode delamination.

To assess the contribution from the passivation layer to preventing ether oxidation, we performed electrolyte exchange studies on the cycled electrodes. After 50 cycles in 1.8 M DPE electrolyte, NCM 811 cathode was harvested from the coin cell and then coupled with 1.8 M DME electrolyte for further cycling (Supplementary Fig. 13). Interestingly, Fig. 3g shows that the known incompatible electrolyte (1.8 M DME) can cycle stably and enables stable CEs with pre-passivated cathode compared to the pristine cathode (99.53% vs. 98.54%). However, the contribution of CEI still cannot bring it to the same stability level of DPE results (99.92%). Such offset is confirmed with the DME-HCE system (Supplementary Fig. 14), where the DME-HCE cathode passivation also cannot completely block dilute ether oxidation (CE shifting from 99.71 to 99.34% after exchange). The results highlight the critical role of a robust cathode passivation. We also demonstrated the minimum CEI solubility into the electrolyte with the enriched FSI⁻ decomposition products, which favors long-term cell cycling (Supplementary Fig. 15 and Note 4). But they also suggest that the surface layer has more prominent kinetic suppression on the reactions that in situ yield the CEI. The improved anodic stability of the DPE electrolyte on the NCM cathode should be a result of synergy between CEI and solvation structure, whose degradation yields the surface passivation. In other words, the stability of DPE electrolyte with abundant free molecules cannot be simply attributed to passivation. After all, if the surface layer could largely block DPE degradation, free DME molecules should also be protected after electrolyte exchange. Additionally, the question still exists of how DPE electrolyte could form an anion-derived CEI despite the enriched free ethers, which are regarded as the less-stable specie. Why are the DPE molecules not preferentially decomposed in the first place, just like the polar DME and DIG systems[39]?

## Electrolyte oxidation behavior and kinetically stabilized interface

Resorting to DFT analyses, we studied the decomposition behaviors of four electrolytes in terms of the electrochemical stability of their solvation structures. Existing DFT studies tend to explain the stability of ether electrolytes via the lowest unoccupied molecular orbital (LUMO) and highest occupied molecular orbital (HOMO) energy levels of individual electrolyte components or coordination couples based on simple solvation models[40]. However, complicated coordination scenarios usually change the stability of the electrolyte species, leading to significant offsets[41]. For the anodic behaviors, solvents coordinating to Li⁺ are usually characterized by higher stability due to a lower HOMO energy level[39,40]. Such phenomenon is fundamental in developing HCEs, since the oxidation stability can be improved when most of the free molecules get coordinated[39]. In our case, we directly extracted the dominant solvation structures in the electrolytes from MD simulations to compute the HOMO/LUMO energy levels, instead of relying on chemical intuition to build possible initial structures. Therefore, the results reflect the oxidation stability and degradation behaviors within different systems. Figure 4a presents the LUMO and HOMO energies of free solvents as well as their corresponding top two, most-probable

solvation structures from MD simulations. The studied ether molecules have very close HOMO energy levels (within 0.1 eV), suggesting their comparable oxidation stability. Therefore, the oxidation stability of the individual solvent molecules cannot explain the improved stabilities in the DPE and DEE electrolytes. However, stark difference originates from the relative positions of the energy levels of the solvation structures compared to the solvents. The strongly anion-involved solvation structures (AGGs) in WSEEs especially in the DPE electrolyte bring their HOMO levels above that of free solvents, resulting in their preferential degradation. As Raman spectroscopy and MD simulation results suggest that Li⁺-FSI⁻ AGG structures prevail over other solvation structures, its decomposition should dominate passivation reactions during the CEI formation process, consequently forming an inorganic LiF-rich CEI and finally leading to complete cathode passivation. Therefore, the reaction equilibrium shifts to the left so that the entire electrode|electrolyte interphase is kinetically stabilized. On the other hand, dilute DME and DIG electrolytes fail to stabilize, since ether molecules are competitively coordinated to Li⁺ but only in small fractions[37]. The majority of ethers are free in the solution and exhibit slightly higher HOMO levels compared to their corresponding solvation structures, and are thus more prone to oxidation[42]. However, these oxidation reactions yield organic products, as demonstrated in our XPS results, that cannot passivate the cathode reaction sites to prohibit continuous oxidation.

In the meantime, the AGG-enriched solvation in the DPE electrolyte also indicates an additional contribution from the interfacial behaviors of ionic clusters[43]. In DPE, the anion preferentially makes the Li⁺ coordination including the released Li⁺ from the cathode during battery charge owing to solvent's low coordination power. Also considering the positively charged cathode and its attraction to FSI⁻, Li⁺-anion aggregations should occupy the surface, repulsing free ether molecules and preventing their direct contact with the cathode surface, which can further mitigate their oxidation[44,45]. To demonstrate the hypothesis, interfacial MD simulations were performed on the cathode surface to determine the ion and solvent distribution in the electrochemical double layer (EDL). Comparing the polar DME electrolyte with the nonpolar DPE electrolyte in Fig. 4b–d, the amount of ether molecules in the EDL of DPE is diminished, which is beneficial for suppressing the direct oxidation of ethers on cathode surface. Meanwhile, the DPE electrolyte also maintains the ion aggregation behavior in the EDL that Li⁺ is coordinated by multiple FSI⁻, as evident from the snapshot in Fig. 4c. Its preferential decomposition leads to the formation of the anion-derived CEI layer as demonstrated in the prior sections. The rearrangement of EDL structure can be realized with the favorable fluidic properties of DPE. Its low viscosity and surface tension can facilitate the migration of the anion aggregates onto the cathode surface, repelling the free ether molecules. In the opposite case, DME solvent molecules are enriched in the EDL over the entire studied distance range. Since most of the DME molecules are uncoordinated to Li⁺ due to the low concentration and cation repulsion effect on the cathode surface, they are very susceptible to the oxidation. The results can also interpret the phenomenon from the electrolyte exchange studies. Despite the well-formed CEI layer via cycling in the DPE electrolyte, the intrinsic solvent-rich EDL structure of the DME electrolyte failed to prohibit the ether oxidation so that the CE of pristine DPE electrolyte could not be reproduced after the exchange. However, the CEI originated from the DPE electrolyte still demonstrated its contribution to protect the electrolyte by increasing the CE by 0.99%. Therefore, the results confirmed the synergistic effect between the solvation structure of the electrolyte and the corresponding CEI in improving the anodic stability, as summarized in Fig. 4e, f.

Finally, it should be noted that stabilizing the NCM811 cathode in the DPE or DEE electrolytes is considered as a kinetic approach, as the oxidation stabilities of the solvents are hardly improved with less

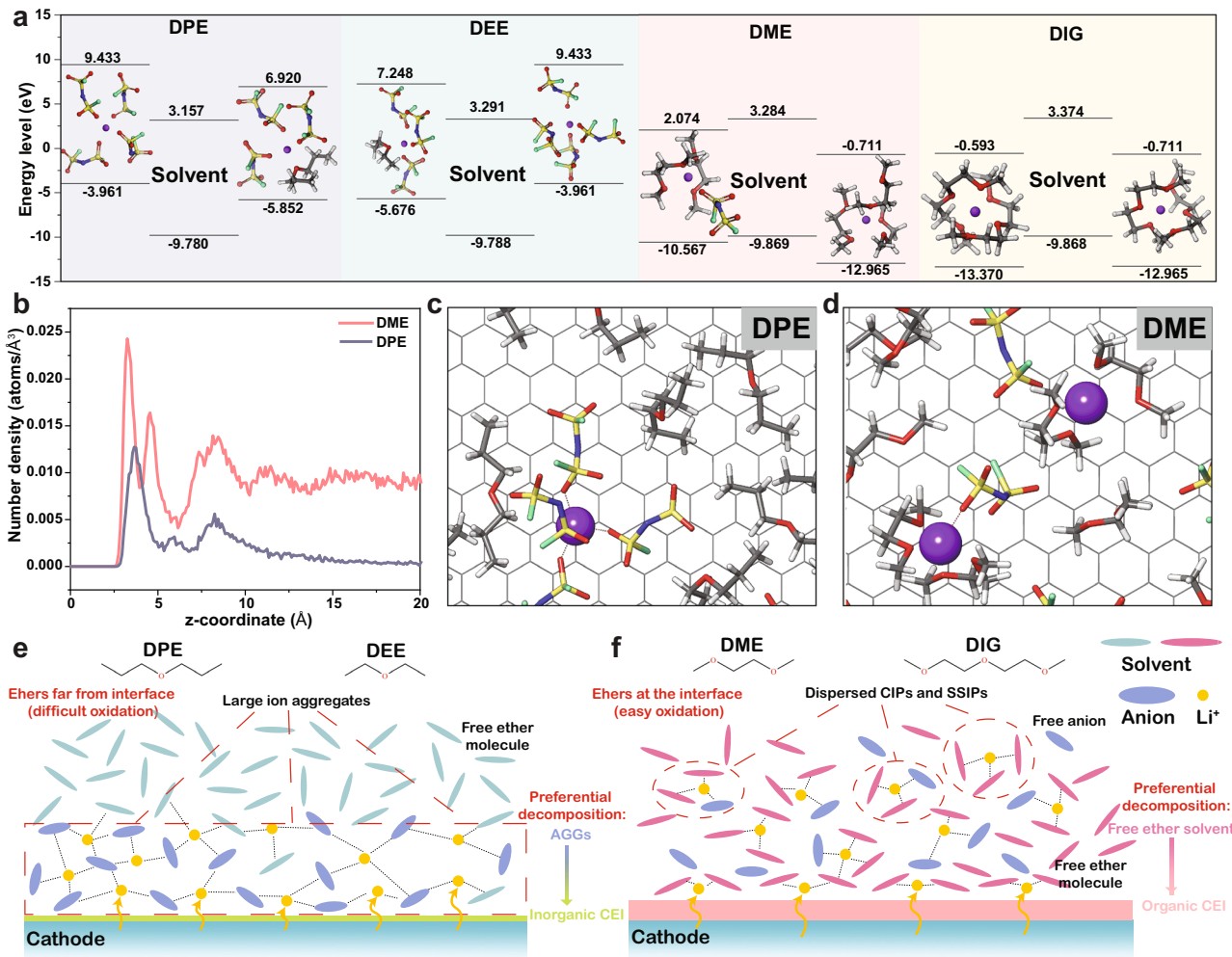

**Fig. 4 | Theoretical study of electrolyte stability and the interfacial models.** All the studied electrolytes contain 1.8 M LiFSI as the salt. **a** Calculated LUMO-HOMO energy levels of solvent and top-two most probable solvation species in different electrolytes. **b** The number densities of ether solvent molecules in the inner Helmholtz layer under applied potential via MD simulation. The DPE (**c**) and DME (**d**) electrolyte components in the EDL at the cathode surface. Illustrations of interfacial model of WSEEs (**e**) and polar ether electrolytes (**f**) at the NCM cathode surface.

coordination to Li⁺ than in the DME or DIG systems. Figure 4a shows that the primary species in the DPE and DEE electrolytes have similar or higher HOMO energy levels than in the DME and DIG systems, indicating their lower thermodynamic oxidation stability. The four studied electrolytes also exhibit similar degradation potential on the inert Pt electrode, which is even close to 1 M LiFSI in DME (but with different decomposition pathway and products, Supplementary Fig. 16, Note 5). However, as demonstrated in our case, unimproved thermodynamic stability does not necessarily lead to incompatibility with high-voltage cathodes and detrimental parasitic reactions. In fact, 1.8 M DPE electrolyte exhibits higher cathode CE than 4.8 M DME, as discussed in the previous sections. Therefore, we have demonstrated a very effective approach to enable high-voltage operation from dilute ether electrolyte, also considering other performance in preventing the Al current collector corrosion and transition metal dissolution (Supplementary Figs. 17, 18 and Notes 6, 7)[10,46].

## Compatibility with Li metal anode
Reversible Li metal deposition and stripping processes are critical in facilitating the stable cycling of LMBs. We thus investigated the impact of the solvation structures. Using the modified Aurbach method[47], Fig. 5a and Supplementary Fig. 19 display the CE of LMA, where 99.42 and 99.47% are determined from DPE and DEE electrolytes, respectively. With stronger solvation power, the DME electrolyte exhibits less

efficiency of 97.56%, while the DIG electrolyte is unsuitable in stabilizing the anode interphase with 36.71% efficiency[7,29]. The results demonstrate the better Li metal compatibilities of DPE and DEE electrolytes, which is also comparable to the efficiency of other state-of-the-art electrolytes[22]. The DME electrolyte shows lower overpotential (-16 mV) than DPE and DEE (-26 mV), mainly due to its ionic conductivity[23]. Long cycle stability test of Li metal anode was performed with Li∥Cu cells. As shown in Fig. 5b, DPE and DEE electrolytes successfully demonstrated 300 stable cycles with average efficiency of 99.45% starting from the 100th cycle. In contrast, the other two electrolytes caused cell failure within 60 cycles.

Ex situ XPS studies of the Li anodes with depth-dependent analyses also identified the improved interfacial passivation with nonpolar ether electrolytes. The cycled Li metal electrodes were harvested from the same Li∥NCM811 coin cells which were used for the cathode ex situ XPS and TEM studies in Fig. 3a–f. The LMA from the DPE electrolyte exhibits very consistent elemental concentrations throughout the SEI in Fig. 5c, which suggests homogenous elemental distributions. The SEI also consists much more abundant fluorinated species (-22%) compared to other electrolytes. It indicates the DPE electrolyte largely enables the anion-originated SEI formation. The DME and DIG-based electrolytes, on the other hand, contain more organic carbon species. Fine XPS spectra analyses in Fig. 5d and Supplementary Figs. 20–23 provide more detailed insights regarding the SEI composition. The

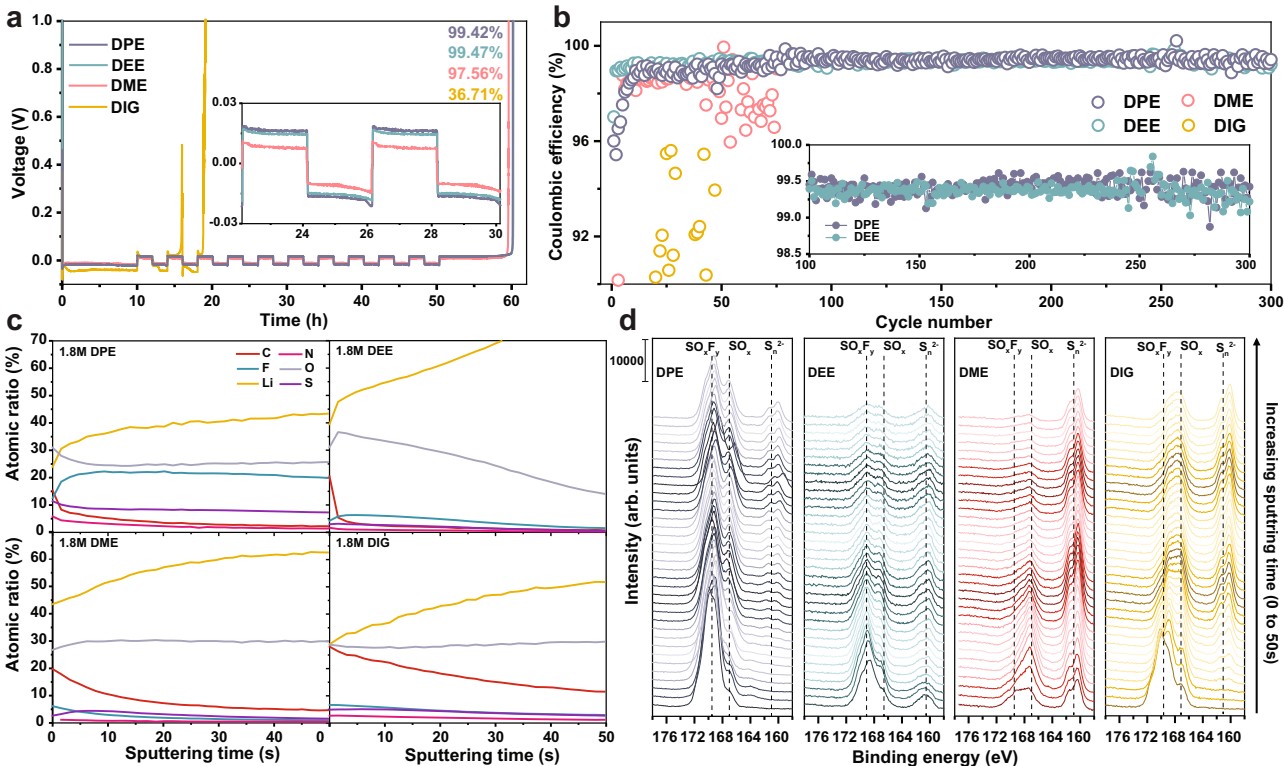

**Fig. 5 | Investigations on the stability of the electrolytes with a Li metal electrode.** All electrolytes contain 1.8 M LiFSI salt. All tests were performed at 25 °C. **a** Coulombic efficiency determined by modified Aurbach method under 0.5 mA/cm² current density. **b** extended long cycling of Li||Cu cells with 0.5 mA/cm², 1mAh/cm². The data points between cycle 100 and 300 are shown in the zoomed-in inset. **c** XPS analyses of depth-dependent elemental concentration on Li metal electrodes. **d** Depth-dependent *S 2p* XPS spectra of cycled LMA from the studied electrolytes. The spectra from bottom to top represent increasing Ar plasma etching depth.

highest intensity of *S 2p* signal from the DPE electrolyte verifies the preferential decomposition of anion and the formation of inorganic species. Abundant $SO_xF_y$ component dominates the reaction products in different depth. The O specie was also mainly detected as S–O components with minimum $Li_2O$. More importantly, chemical compositions are also observed to vary with increasing etching depth from all the studied electrolytes, but to different extents. As etching proceeds, compounds with lower binding energy are identified in the SEI, such as $SO_xF_y \rightarrow SO_x \rightarrow S_n^{2-}$ (*S 2p*); $F-N(SO_2) \rightarrow N(SO_2)$ (*N 1s*); $S-O \rightarrow C-O \rightarrow Li_2O$ (*O 1s*). It indicates more complete reduction occurs as approaches to the metallic Li. Among the four electrolytes, the WSEEs like the DPE and DEE electrolytes feature less SEI component variations in different depth, which improves the Li metal passivation and reduces the parasitic reactions. The Li deposition efficiency can thus be enhanced compared to DME and DIG. Li metal deposition morphology studies via ex situ scanning electron microscopy (SEM) measurements (Supplementary Fig. 24) verified the improved plating behavior when decreasing the solvating power. DPE and DEE achieved uniform Li deposition on Cu substrate with large Li grain size. Their cross sections indicate dense Li packing with minimal cracking and porous structures. The DME electrolyte shows ununiform plating where the substrate can be observed. Due to the low LMA CE, the DIG electrolyte hardly plated Li and generated a very thin layer of deposition with dendritic morphology. Improved Li deposition behavior is also due to demonstrated better wettability on the Li metal surface (Supplementary Fig. 25).

### Testing of the ether-based electrolyte solutions in Li||NCM811 cell configuration under practical conditions

Lastly, we investigated the electrochemical performance of the four electrolytes within practical LMBs under controlled electrolyte and Li anode amount conditions. High loading NCM811-based positive electrodes (-16.5 mg/cm², 3.3 mAh/cm²) were used in the following studies with 4.3 V high cutoff voltage. To accurately control the Li metal amount, electrochemically deposited 2-times excess Li on Cu (N/P ratio = 2) were used as negative electrode in coin cell configuration. Firstly, Fig. 6a shows the prolonged cycling stability of LMB with the DPE electrolyte compared to the other three electrolytes. The DPE electrolyte retained 82% capacity after 220 cycles, whereas DME and DIG caused quick battery failure before 100 cycles (Supplementary Fig. 26). The obtained average CE of DPE cell was 99.90%, indicating the well-stabilized cathode electrolyte interface with mitigated side reactions. The 1.8 M DEE electrolyte also delivers improved cyclability but cannot outcompete DPE due to its suboptimal high-voltage compatibility, as shown in the previous sections. With the advantage of weak coordination to Li⁺, the DPE and DEE electrolytes facilitate stable cycling at 3.3 mA/cm² (Fig. 6b and Supplementary Fig. 27). The DME and DIG cells, instead, failed to complete the rate capability test because their poor plating/stripping efficiency caused quick Li anode consumption at high current densities. Using 200 μm Li anode for the DME and DIG-based electrolytes managed to obtain their rate performance, while they still show faster capacity decrease and battery failure despite the excessive amount of Li. However, it can be noticed the ionic transport limitation for Li||NCM811 cell with high-mass-loading positive electrodes. The DPE requires large overpotential at 3.3 mA/cm², where around 40% of capacity were charged during the constant-voltage step (Supplementary Fig. 27). In contrast, the DME electrolyte with the better ionic conductivity demonstrates the lowest overpotential. Therefore, our results verify that the rate performance is governed by both electrolyte ionic conductivity and the interfacial behaviors[48]. They are supposed to be modulated simultaneously for the optimized high-rate battery performance. The DPE or DEE electrolytes which features fast interfacial kinetic and high CE also suffer from limited bulk ionic transport due to the strong Li⁺-anion

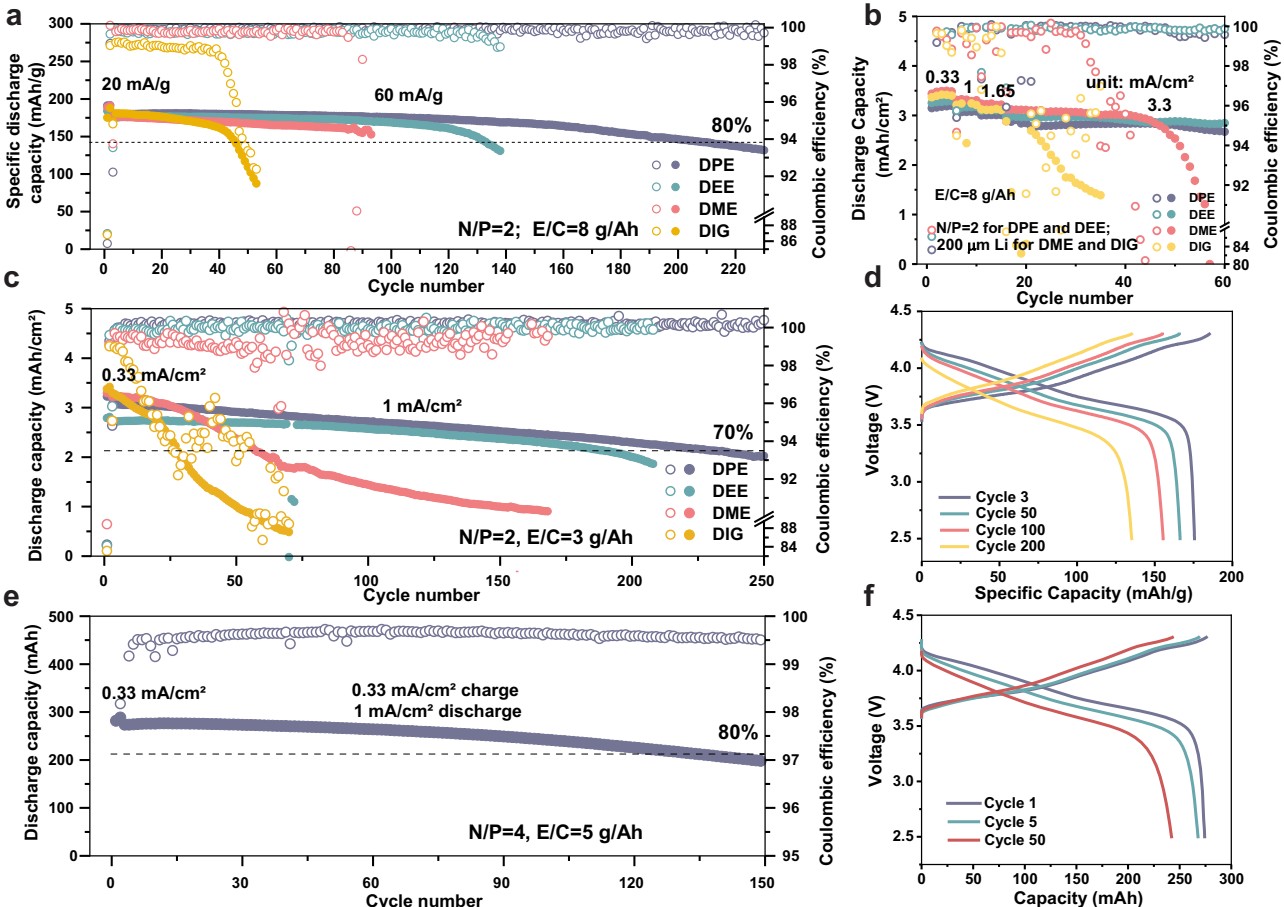

**Fig. 6 | Electrochemical testing of the electrolyte solutions in Li‖NCM811 coin and pouch cell configurations.** All tests were performed at 25 °C with Li metal negative electrode, NCM811 positive electrode (3.3 mAh/cm², 16.5 mg/cm²), and each electrolyte contains 1.8 M LiFSI salt. **a** Long cycling study of the coin cells with N/P ratio = 2, E/C ratio = 8 g/Ah. Two cycles were performed at 20 mA/g, followed by long-term cycling at 60 mA/g. The mass of the specific current and specific capacity refers to the mass of the active material in the positive electrode. **b** Rate study of the coin cells with different electrolytes at 0.33, 1, 1.65, and 3.3 mA/cm². **c, d** Lean electrolyte condition study of coin cells with N/P ratio = 2, E/C ratio = 3 g/Ah and the voltage profiles of DPE electrolyte. The cells were cycled at 0.33 mA/cm² for two cycles followed by long-term cycling with 0.33 mA/cm² charge and 1 mA/cm² discharge. **e, f** Long cycle demonstration and cycling profiles of the DPE-electrolyte-based 300 mAh double-sided pouch cell under 30 PSI pressure. The pouch cell was cycled at 30 mA for two times followed by long-term cycling at 30 mA charge, 90 mA discharge.

interactions. The DPE electrolyte also enabled long cycling of LMB under lean electrolyte condition with the E/C ratio of 3 g/Ah. The low viscosity and surface tension of the DPE electrolyte (Fig. 1b) are beneficial for wetting the separator/electrodes and the penetration into the microstructures despite the restricted electrolyte amount[49,50]. Under such a strict scenario which mimics the real LMB operation[37], the DPE electrolyte still demonstrated an 80% capacity retention after 172 cycles, shown in the Fig. 6c, d. Additionally, a graphite‖NCM811 coin cell tested with the 1.8 M DPE-based electrolyte delivered long-term 1000 stable cycles with an average CE of 99.95% (Supplementary Figs. 28–30 and Notes 8–10). Other testing conditions including low temperature[51] and anode-free cell (Supplementary Figs. 31, 32) also demonstrated the improved performance from the nonpolar DPE electrolyte. Finally, we assembled 300 mAh Li‖NCM811 pouch cell prototypes with the DPE electrolyte to assess its performance in large-size batteries with moderate stacking pressure (around 30 PSI, 200 KPa) (Supplementary Fig. 33). As shown in Fig. 6e, f, the pouch cell demonstrated a capacity retention of 88.6 and 74.1% after 100 and 150 cycles, respectively.

## Discussion

In summary, we demonstrated a strategy for improving high-voltage compatibility of dilute (<3 M) ether-based lithium electrolytes by using the highly nonpolar ether solvent. Our low concentration LiFSI-based electrolyte with nonfluorinated DPE solvent was shown to extend the high voltage (4.3 V) operation of LMBs with commercially viable battery configurations (high loading cathode, controlled anode, and electrolyte amount). Due to the weak coordination ability to Li⁺ and the ion aggregation enriched Li solvation behavior, DPE tunes the relative HOMO energy level of aggregated solvate species and rearranges the decomposition order of electrolyte components at cathode surface. The preferential degradation of ion aggregates circumvents the oxidation of free ether molecules and leads to a robust anion-derived CEI layer. The aggregated Li solvation structure displaces the ether molecules in the EDL, leading to solvent-deficient interfacial regime and synergistically enhances ion transfer process. Compared to the conventional electrolyte design strategies, stabilizing the positive electrode interface enables a CE of 99.90% by using the DPE electrolyte does not require diminishing the free ether molecules or improving the thermodynamic stability of the electrolyte. The Li‖NCM811 coin cell retains 82% capacity after 220 cycles at 1 mA/cm², and the practical pouch cell also demonstrates 150 stable cycles with 74.1% retention (0.33 and 1 mA/cm² charge and discharge, respectively). This study suggests a way to overcome the poor solvent stability of ether-based electrolytes with high-voltage cathodes by modulating the locally controlled and dynamically changing solvation structure. We

established an effective electrolyte design that can construct kinetically controlled interface to enhance LMB operations. Our results also provide insights of understanding the CEI formation process and tuning its chemical compositions.

## Methods

### Materials

Anhydrous diglyme (99.5%), 1, 2-Dimethoxyethane (99.5%), diethyl ether (99.5%) were purchased from Sigma Aldrich and used without any treatment. Dipropyl ether (99%) from Sigma Aldrich was dried with 4 Å molecular sieves for one week in the Ar-filled glovebox at 25 °C before use. Lithium bis(fluorosulfonyl) imide salt (99.5%) was purchased from Solvionic and used without any treatment. The separator used in this study was Celgard 2500 (25 μm, 0.55 porosity, 2.5 ± 0.2 tortuosity[49]). The NCM811 electrodes with ~1.6 mAh/cm$^2$ (37.3% porosity, calendered thickness of 52 μm, and area loading of 8.21 mg/cm$^2$) were provided by Argonne Nation Laboratory. The high loading NCM811 electrode (~3.3 mAh/cm$^2$, 16.5 g/cm$^2$) and the corresponding graphite anode were provided by Battery Innovation Center. For the NMC811 cathode, the as-coated density/porosity was 2.40–2.42 g/mL, porosity of 45.0–45.5%. It was then calendered to a thickness of 127–128 μm, corresponding to a density of 3.04–3.15 g/mL (variation due to tolerances in mass loading), and a porosity of 28.4–31.0%. It was densified by passing through an 80 °C calender. The graphite anode had an as-coated porosity/density of 51%/1.05 g/mL. It was then calendered to a density of 1.42–1.47 g/mL, and a calculated porosity of 32–33%. The anode was also densified via calendering at 80 °C. The calender machine was manufactured by Independent Machine Co.

### Electrochemical characterization

All the electrochemical characterizations with 2032-type coin cells except Li-metal anode studies were performed with Al-clad cathode lids purchased from MTI corp. All the coin cell preparations were carried out in a Ar-filled glovebox. The pouch cell was prepared in a dry room.

Li||NCM811 (1.6 mAh/cm$^2$) coin cells were prepared with Ø 12 mm cathode, 40 μl electrolyte solution, Celgard PP separator (Ø 19 mm), and Ø 14 mm lithium disk (99.9% pure lithium metal from Sigma-Aldrich). One 19 mm Al foil (15 μm, >99.3% purity, MTI Corp.) was placed in the cathode lid to further prevent stainless steel corrosion. The Li metal cells were pre-cycled twice between 2.5 V and 4.3 V at 0.16 mA/cm$^2$ rate and further long-cycled at 0.48 mA/cm$^2$ rate with constant voltage holding at 4.3 V until the current dropped to 0.16 mA/cm$^2$. The specific capacities were calculated based on the weight of active NCM811 material. The 3.3 mAh/cm$^2$ NCM811 cathodes were used for the Li metal battery studies under practical conditions. For coin cells, Li metal anodes were prepared by electrochemically depositing 7 mAh/cm$^2$ Li metal on Cu foil (9 μm thick, Purity > 99.8%, MTI Corp.) with the current density of 0.5 mA/cm$^2$. Coin cells with the DIG-based electrolyte used 50 μm Li foil as anode since it cannot achieve reversible Li plating. The sandwich-type pouch cells were prepared with a single piece of double-sided cathode (6.8 mm *6.8 mm) and two pieces of 50 μm Li foil on Cu substrate (7 mm* 7 mm) in a dry room with the dew point at around −60 °C. 1.5 g of DPE electrolyte was added. The designed pouch cell fixture was applied with 30 PSI pressure (Supplementary Fig. 33c). The graphite||NCM811 coin cell was prepared with Ø 12 mm cathode (3.3 mA/cm$^2$), Ø 12.7 mm graphite anode with the N/P ratio of 1.05, and 40 μl electrolyte solution. The anode-free Cu||NCM811 coin cell was prepared with Ø 12 mm cathode (3.3 mA/cm$^2$), Ø 16 mm Cu foil as the anode, and 40 μl electrolyte solution.

Linear scanning voltammetry studies were performed within the Li||Al coin cells with the Gamry Potentiostat at a scan rate of 0.5 mV/s. The Current-Interrupt IR Compensation was applied. To control the electrolyte amount and regulate the reaction area, controlled 40 μL of electrolytes and glass fiber separators were used during asymmetric cell assembly. The bulk electrolyte ionic conductivity tests were performed at 25 °C with a Metrohm Autolab potentiostat instrument equipped with a FRA32M module and an Autolab Mircrocell high temperature (HT) cell. 1 mL of each electrolyte was added to the cell and tested.

For the electrolyte exchange study (Supplementary Fig. 13), the primary Li||NCM811 (1.6 mA/cm$^2$) coin cell was cycled 50 times with the 1.8 M DPE electrolyte at 0.48 mA/cm$^2$. In the Ar-filled glovebox, the cathode and the extra piece of Al were harvested from the original cell and rinsed with DPE solvent. After vacuum drying, the combination was assembled into a new cell but with 1.8 M LiFSI DME electrolyte. The extra Al foil was assembled in the cell and transferred together with the cathode in order to avoid the fresh Al-clad cathode casing from the exchanged cell being exposed to the DME-based electrolyte and cause additional corrosion. 40 μl electrolytes were applied to the original and exchanged cell.

Li metal anode studies were performed in Li||Cu coin cells with 16 mm Cu foil and 14 mm Li metal disk (200 μm thick). The Cu foil was sonicated for 10 min with water and ethanol and then vacuum dried at 80 °C before use. The coulombic efficiency value of the Li metal anode was measured by the modified Aurbach method in the Li||Cu coin cells with current density of 0.5 mA/cm$^2$ after one formation cycle (0.5 mA/cm$^2$, 5 mAh/cm$^2$). The coulombic efficiency was determined by:

$$CE = \frac{nQ_c + Q_s}{nQ_c + Q_T} \qquad (1)$$

Where $n = 10$, $Q_c = 1$ mAh/cm$^2$, $Q_T = 5$ mAh cm$^2$, $Q_S$ is the specific area capacity of the final charge step.

All the electrochemical measurements except the low/high temperature tests were carried out in the lab at 25 °C without environmental chambers. The temperature dependent study was performed in the ESPEC environmental chamber from −40 to 50 °C. At least two cells were measured for each electrochemical test.

### Material characterization

Raman spectroscopy studies were performed with Thermo Fisher Scientific DXR Raman microscope with a 633 nm laser source. The Raman microscope has the objective magnification of 50× and the grating of 600 lines/mm. The spectral resolution of the collected Raman data is 1 cm$^{-1}$. 7.5 mW laser power, 7.5 s exposure duration, and 50 scans were used for Raman spectra acquisition (Supplementary Fig. 34 and Note 11).

The surface tension of the electrolyte solutions and solvents were tested at 25 °C by the pendant-drop method with the Advanced Goniometer from Rame-hart Instrument Co. The contact angle testing was performed at 25 °C in the Ar-filled glove box. The pictures were taken through the glovebox glass shield 5 seconds after dropping 15 μL of different electrolyte solutions onto the Li metal surface. The viscosity and density were tested at 25 °C by the Rolling-ball viscometer Lovis 2000 from Anton Paar.

SEM imaging was conducted with Teneo Volumescope. 4 mAh/cm$^2$ Li metal samples were electrochemically deposited on Cu substrate in Li||Cu coin cells at 0.5 mA/cm$^2$. The coin cells were then opened in the Ar-filled glove box. They were rinsed with corresponding electrolyte solvent and vacuum dried at 25 °C. The samples were then loaded onto the SEM holder in the glovebox and transferred with sealed, Ar-filled sample vials. The loaded SEM holders were quickly taken out from the vials and transferred to the instrument before imaging. The samples were exposed to air less than 5 s. The EDS spectra of cycled Li anode was collected with Thermo Scientific Helios G4 UX Dual Beam electron microscope.

[7]Li and [19]F NMR spectra at 25 °C were obtained on a 500-MHz liquid NMR spectrometer (Agilent, USA) with a 5 mm HX probe at the Larmor frequencies of $2\pi \times 233.23$ and $2\pi \times 470.59$ rad·MHz/T using single pulse excitation. The 0 ppm chemical shifts of [7]Li and [19]F were calibrated with external references, 1 M LiCl solution and 2-2-2-Trifluoroethanol ($CF_3CH_2OH$), respectively.

Thin samples for transmission electron microscopy (TEM) were prepared from the cycles cathode material using the focused ion beam (FIB) lift-out technique in Helios Hydra UX dual-beam plasma FIB/scanning electron microscope (PFIB). The high-resolution phase-contrast TEM images depicting the atomic fringes at the CEI layers on cycled cathodes were captured using an aberration-corrected Thermo Fisher Scientific environmental transmission electron microscope (ETEM) operated at 300 kV.

X-ray photoelectron spectroscopy (XPS) was used to measure the surface composition and chemical state of elements in as received and oxidized samples as a function of depth. After the cell opening in the glovebox, the cycled cathode and anode were rinsed in 1 mL of the corresponding fresh ether solvent for 10 min and 3 times (Supplementary Fig. 35 and Note 12). The electrodes were vacuum dried at 25 °C in the glovebox and then sealed under Ar atmosphere into the sample vials before transferring into the XPS sample preparation glovebox to avoid samples exposing to air. XPS measurements were performed using a Nexsa ThermoFisher Scientific spectrometer, using focused Al Kα monochromatic X-ray source (1486.6 eV) operated at 72 W and a high-resolution spherical mirror analyzer using 50 eV pass energy. The data acquisition was carried out using a 300 μm diameter X-ray beam and emitted photoelectrons were collected at the analyzer entrance slit normal to the sample surface. The chamber pressure was maintained at $-5 \times 10^{-9}$ Torr during the measurements. All the XPS peaks were charge referenced using *C 1s* binding energy of 284.8 eV. XPS data were analyzed by CasaXPS software using Shirley background correction. For depth profile measurement, 300 eV mono energetic argon ions were used. The ion beam was rastered about 1.2 mm × 1.2 mm area and the XPS measurements were performed at the center of the crater using 300 μm diameter X-ray beam. The Sputter depth scale was calibrated using $Ta_2O_5$ sputter rates.

### Computational methods

**MD simulations.** The initial configurations of the four electrolyte systems were first built using the PACKMOL[52] package by randomly placing the species in a $5 \times 5 \times 5$ nm³ cubic box. The OPLS-2005[53] force field of Schrödinger (provided by Schrödinger, LLC, New York, NY, 2021) was used for all solvent molecules, while the Lopes-Pádua[54] force field parameters were used for FSI⁻. Li+ parameters were taken from the work of Dang et al.[55] Electronic polarizability effects were accounted for implicitly by using the electronic continuum correction (ECC) method[56,57], in which partial atomic charges of the ionic species were replaced by effective charges using the electronic high-frequency dielectric permittivity of the solvent. MD simulations were performed using the LAMMPS[58] (https://www.lammps.org) open-source software version 3 Mar 2020. Lennard Jones interactions were truncated at a cutoff distance of 1.2 nm and the particle-particle particle-mesh (PPPM)[59] method was used for long-range electrostatic interactions. The prepared systems were first minimized using the steepest descent with a convergence criterion of 1000 kcal/mol Å followed by conjugated gradient with a convergence criterion of 10 kcal/mol Å. The systems were then equilibrated to a temperature of 298.15 K and a pressure of 1 atm in the NPT ensemble with a time constant of 0.001 ps for 2 ns. Equilibrated systems were then melted to 500.15 K for 2 ns and then quenched to 298.15 K in four steps for 3 ns. Lastly, production runs in the NVT ensemble at 298.15 K were performed for 5 ns using a timestep of 0.001 ps, from which properties of interest were derived. All MD simulations and analysis and DFT calculations were run using our automated high-throughput infrastructure (MISPR)[60].

To simulate the electrode-electrolyte interface, the equilibrated DME and DPE electrolyte systems were added between two graphene slabs with opposite net charges to represent the positive and negative electrodes (Supplementary Note 13). The OPLS-AA force field was adapted for the graphene slabs with corrected atomic charges to mimic an applied electrode potential. We note that this constant charge method has been found appropriate for modeling the behavior at electrode–electrolyte systems like the ones reported in this work[61]. A layer of vacuum of 10 nm was wadded in the *z*-direction to separate periodic images and reduce Columbic interactions between the mirrored slabs. Following that, an NVE equilibration was performed for 2 ns using the Langevin thermostat, followed by an NVT production run at 298.15 K for 5 ns, from which number density profiles were extracted and analyzed.

**DFT calculations.** DFT calculations were performed using Gaussian 16 Rev.C.01 (calculation software provided by Gaussian, Inc., Wallingford, CT, 2016). Binding energy and HOMO/LUMO energy levels calculations for the individual molecules and the top solvation structures identified from the MD simulations in each electrolyte system were performed at the ωB97X[61]/def2-TZVP level of theory with explicit solvent molecules. The geometry of the molecules and complexes were first optimized to obtain accurate structures. A vibrational frequency analysis was then carried out at the same level of theory following the optimization of each structure to ensure the optimized geometry corresponds to a minimum. The binding energy, in kJ/mol, was calculated as $E_{complex} - (E_{Li^+} + nE_{solvent} + mE_{anion})$, where $E$ is the energy and $n$ and $m$ are the number of solvent and anion species in the cation solvation shell, respectively.

## Data availability

The data that support the plots within this paper and other finding of this study are available from the corresponding author upon reasonable request.

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

## Acknowledgements

V.G.P. and Z.L. truly thank the Program Manager, Maria Medeiros for the financial support from the Naval Enterprise Partnership Teaming with Universities for National Excellence (NEPTUNE), Office of Naval Research (Grant # N000142112070) and NEPTUNE-N00014-22-1-2333. The 1.6 mAh/cm$^2$ NCM811 electrode was obtained from the U.S. Department of Energy's (DOE) CAMP (Cell Analysis, Modeling, and Prototyping) Facility, Argonne National Laboratory. V.G.P. also thanks DoD's Defense University Research Instrumentation Program (DURIP) for the project 'Li-ion Battery Safety Systems: In situ/Multi-mode Calorimetry, Electrochemical Impedance Spectroscopy, and Critical Temperature Cycling', which was used to establish 'Ultralow Temperature ViPER Batteries (#N00014-22-1-2333) Laboratory at Purdue's ChE school. Z.L. thanks Sotoudeh Sedaghat from Materials Engineering department, Purdue University for the help on EDS study. NMR analysis was supported by the Joint Center for Energy Storage Research (JCESR), an Energy Innovation Hub funded by the U.S. Department of Energy (DOE), Office of Science, Office of Basic Sciences (BES). NMR experiments were performed in the Environmental Molecular Sciences Laboratory (EMSL), a national scientific user facility sponsored by the Department of Energy's Office of Biological and Environmental Research located at Pacific Northwest National Laboratory (PNNL). High-performance computational resources for this research were provided by the Stony Brook Institute for Advanced Computational Science (IACS). N.N.R. was supported by the startup funds from Stony Brook University. R.A. acknowledges the IACS Junior Research Award.

## Author contributions

Z.L. conceived the idea and designed the project. V.G.P. supervised the project. Z.L. and H.R. conducted electrolyte preparation, electrochemical testing, and material characterization. R.A. performed the MD simulations and DFT calculations. B.M.S. performed the XPS spectra acquisition and analyzed the data with Z.L. The pouch cell fabrication and studies were conducted by S.G. The HR-TEM and NMR characterization results were obtained by B.G., T.A.A., and K.S.H., respectively. T.A.E. obtained the contact angle and surface tension results. V.G.P., N.N.R., and V.M. provided important insights during the project design and supervision. Z.L. wrote the paper, with contributions from all authors.

## Competing interests

The authors declare no competing interests.
