## [Peer Review File · Nature Communications]

REVIEWER COMMENTS

Reviewer #1 (Remarks to the Author):

Review report for manuscript:

Nature Communications manuscript NCOMMS-22-29400-T

The article by Zheng Li, Vilas Pol et al. that I received for revision presents an effective strategy to mitigate batteries performance fading by a cleverly selecting the electrolyte according to its Li-ion solvation structure and dielectric constant. I found the approach sound and convincing and the methodology of scientific analysis robust and exhaustive. The results discussed in this paper have implication in different energy storage devices based on different ion chemistries, and - generally speaking – highlight the importance of carefully considering ion solvation structures and solid-liquid interactions at electrified interfaces as key aspects to improve device performances and extend devices lifespan. I believe this broader scope implications should be emphasized.

I would recommend this work to be published in Nature Communications after minor revisions. Here below a list of suggestions that might help to clarify some aspects of the research.

1. The authors introduce the different electrolytes considered classifying the according on their dielectric constant. Whilst it is true that this property affects Electrical double layer (EDL) stability at the electrode-electrolyte interface, I believe that also fluidic properties (particularly viscosity and Surface tension) should be considered (as the authors suggest in Supplementary Figure 22). It would be beneficial to integrate this information in the Figure 1 (or adding a Supplementary Figure). Also, the effect on solvents viscosity given by the addition of Li salts can help to better argument the sections discussed at page 13 lines 1-5. (see ref. 1)
2. The authors demonstrate and discuss the effect of different dielectric constant on LMB NMC811/Li chemistries and half cell. They also demonstrate a practical NMC811/Graphite full cell (Supplementary Figure 23), however I believe that the implications of this approach - i.e. tailoring the solvation strength of solvents - on Negative electrodes of NMC 811/Gr practical full cells should also be further discussed (Page 16, line 21 – please note missing space) and framed in other degradation mechanisms, for instance electrode slippage. (see ref. 2)

3. Supplementary information – Materials (Lines 5-6): More details on Positive electrode processing and parameters (porosity/tortuosity, calendaring) could be provided. (see refs. 3,4)

A discussion on electrolyte invasion in the electrode depending on surface tension/viscosity and electrodes exposed surface could be added in the main text.

4. Do the authors apply any surface treatment/washing to the Cu foil before Li-metal is electrodeposited and / or to electrodeposited Li?

5. Which are the properties (Porosity, tortuosity, surface coating) of the separators used? Do the authors believe that the choice of solvent might have an effect on the (local) microviscosity and thus on the Li-ion transport properties / solvation of the electrolyte? (see refs. 1,5,6)

6. Page 15, line 27 (supplementary Figure 21): can the authors provide (at least an approximate) values of contact angle – I suppose the pics were taken in glovebox atmosphere. The conditions (Temperature, pressure, atmosphere) could be specified in caption.

7. The experimental section (Supplementary Information page 1, lines 29-30) describing the Raman spectroscopy analysis in Figure 2e (main text) should include the objective magnification, spectral resolution and the grating used for the spectrographic analysis. Do the authors perform the analysis at different pump power to exclude any thermal effects?

Recent literature on operando Raman analysis of battery electrolytes suggested that changes in the electrolyte solvation structure are correlated to the SEI formation. (refs. 7 and 8) These reports could support the claims of the authors and should be cited.

Minor:

- Figure 1c , inset: hard to read.

- Page 3, line 20: coma missing between 8 and 10 (8,10)

- Page 5, line 4: a citation to support the steric hindrance argument would be beneficial.

References:

1. Sun, Y., Radke, C. J., McCloskey, B. D. & Prausnitz, J. M. Wetting behavior of four polar organic solvents containing one of three lithium salts on a lithium-ion-battery separator. *Journal of Colloid and Interface Science* 529, 582–587 (2018).
2. Dose, W. M., Xu, C., Grey, C. P. & De Volder, M. F. L. Effect of Anode Slippage on Cathode Cutoff Potential and Degradation Mechanisms in Ni-Rich Li-Ion Batteries. *Cell Reports Physical Science* 1, 100253 (2020).
3. Sauter, C., Zahn, R. & Wood, V. Understanding Electrolyte Infilling of Lithium Ion Batteries. *J. Electrochem. Soc.* 167, 100546 (2020).
4. Landesfeind, J., Hattendorff, J., Ehrl, A., Wall, W. A. & Gasteiger, H. A. Tortuosity Determination of Battery Electrodes and Separators by Impedance Spectroscopy. *J. Electrochem. Soc.* 163, A1373 (2016).
5. Schilling, A. et al. Influence of Separator Material on Infiltration Rate and Wetting Behavior of Lithium-Ion Batteries. *Energy Technology* 8, 1900078 (2020).
6. Saito, Y., Takeda, S., Morimura, W., Kuratani, R. & Nishikawa, S. A Selective Interaction between Cation and Separator Membrane in Lithium Secondary Batteries. *J. Phys. Chem. C* 121, 23926–23930 (2017).
7. Miele, E. et al. Hollow-core optical fibre sensors for operando Raman spectroscopy investigation of Li-ion battery liquid electrolytes. *Nature communications* 13, 1–10 (2022).
8. Mozhzhukhina, N. et al. Direct Operando Observation of Double Layer Charging and Early Solid Electrolyte Interphase Formation in Li-Ion Battery Electrolytes. *J. Phys. Chem. Lett.* 11, 4119–4123 (2020).

Reviewer #2 (Remarks to the Author):

The manuscript by Li et al. reports on a novel approach of using highly nonpolar electrolytes for Li/NMC batteries. The data presented demonstrates higher coulombic efficiency for cells with more nonpolar ethers as compared to the more polar ethers used in this study. Electrolytes solvation structure in the bulk and at the interface are explored by several experimental and theoretical methods. Authors also report the effect of electrolyte structure on the electrode/electrolyte interface composition. This work explores the stabilization of metal lithium anode and high voltage cathodes, which is a very relevant

topic. However, the main concern is that electrolytes employing nonpolar solvents have much lower conductivity, that would affect battery performance, which is not discussed in detail in the manuscript.

The following concerns and questions should be addressed by authors before publication is possible:

1. Figure 1b shows electrochemical stability windows of the electrolytes, however DPE and DEE based electrolytes have about 10 times lower conductivity (as reported in supl. Fig. 3). Was ohmic drop IR correction performed in this measurement? Without ohmic drop correction the stability windows estimation is not reliable. Please include IR drop correction in the experimental details and the LSV curves with and without IR correction.
2. The highest C-rate used in the study is C/3, which is a relatively low C-rate for commercial applications. Did authors try to perform higher C-rates measurements, for example 1C? Is C/3 the maximum C-rate possible due to the low electrolyte conductivity? Please add the discussion about performance limitations due to the low electrolyte conductivity.
3. Please add the Raman spectra resolution in the experimental details.
4. Ex situ XPS measurements provide conclusions that less polar electrolyte leads to more inorganic SEI. Please add in the experimental details the procedure for ex situ samples preparation. Were the electrodes washed with the solvent, and in that case with which solvent? Is there any possibility that detected F-rich layer is due to the electrolyte salt residue rather than SEI? If the electrodes were washed with non-polar solvent, it is more probable that there would be salt residues, that could from LiF as a result of argon etching.
5. Could SEI components low solubility in the electrolyte be a contributing factor for the SEI stability? Could this also explain the results of electrolyte exchange studies, considering that SEI components are likely to be more soluble in DME as compared to DPE?
6. The interfacial MD simulations on cathode surface were performed using graphene slab as electrode interface. Could you please comment how comparable and relevant the charged graphene interphase to the NMC cathode, could the nature of interface influence the results?

Major revision is recommended.

Reviewer #1

The article by Zheng Li, Vilas Pol et al. that I received for revision presents an effective strategy to mitigate batteries performance fading by a cleverly selecting the electrolyte according to its Li-ion solvation structure and dielectric constant. I found the approach sound and convincing and the methodology of scientific analysis robust and exhaustive. The results discussed in this paper have implication in different energy storage devices based on different ion chemistries, and generally speaking highlight the importance of carefully considering ion solvation structures and solid-liquid interactions at electrified interfaces as key aspects to improve device performances and extend devices lifespan. I believe this broader scope implications should be emphasized.

I would recommend this work to be published in Nature Communications after minor revisions. Here below a list of suggestions that might help to clarify some aspects of the research.

Comment 1: The authors introduce the different electrolytes considered classifying the according on their dielectric constant. Whilst it is true that this property affects Electrical double layer (EDL) stability at the electrode-electrolyte interface, I believe that also fluidic properties (particularly viscosity and Surface tension) should be considered (as the authors suggest in Supplementary Figure 22). It would be beneficial to integrate this information in the Figure 1 (or adding a Supplementary Figure). Also, the effect on solvents viscosity given by the addition of Li salts can help to better argument the sections discussed at page 13 lines 1-5. (see ref. 1)

Response: We appreciate this scientific comment from the reviewer. We also agree that the fluidic properties of electrolytes can affect their ion transfer and interfacial behaviors. Here, we have measured the surface tension and viscosity values of all the studied solvents and electrolytes. The plot has been integrated in Figure 1b. The detailed surface tension testing plot is included in the Supplementary Figure 2.

Among the studied ethers, monodentate ethers (DPE and DEE) with smaller molecular size exhibit consistently smaller surface tension and viscosity, which favors wetting the porous electrode and separator surface. The wetting behavior study conducted by Sun et al. suggests that the electrolyte surface tension below 26.1 mF/m is optimum for wetting the Celgard 2500 separator¹, which is very well correlated with our results, as the DPE and DEE electrolyte shows excellent wettability on the same separator with low surface tension (22.41 and 22.80 mF/m, respectively).

In terms of the interfacial behavior on the cathode, the rearrangement of ELD structure within the electric field can be easier realized due to the favorable fluidic properties. Therefore, the low solvent viscosity and surface tension of the DPE electrolyte can facilitate the migration of the anion aggregates onto the cathode surface, repelling the free ether molecules.

We added the discussions to the main text (Page 5, line 4; Page 10, line 16) as well as the SI (Page 3). The recommended literature was cited.

Figure 1b. The viscosity and surface tension values of the studied electrolytes and solvents.

Supplementary Figure 2. The surface tension testing results of the studied ether solvents and their corresponding electrolyte.

Comment 2: The authors demonstrate and discuss the effect of different dielectric constant on LMB NMC811/Li chemistries and half cell. They also demonstrate a practical NMC811/Graphite full cell (Supplementary Figure 23), however I believe that the implications of this approach - i.e. tailoring the solvation strength of solvents - on Negative electrodes of NMC 811/Gr practical full cells should also be further discussed (Page 16, line 21 – please note missing space) and framed in other degradation mechanisms, for instance electrode slippage. (see ref. 2)

Response: We thank the reviewer for this thoughtful comment about the impact of ether electrolyte solvation structure on the performance of the graphite anode. In the older manuscript, we decided to include the performance of the NCM811-graphite LIB full cell as evidence to demonstrate the excellent long-term antioxidation capability of the DPE electrolyte, as we obtained ultra-high CE from the high-voltage LIB within 1000 cycles. Here, we agree with the reviewer that it is also meaningful to further elucidate the advantages of using WSEE for the graphite anode to highlight the importance of solvation structure on different energy storage devices. During the paper revision, we tried to justify the outstanding graphite performance with suppressed electrode slippage in the DPE electrolyte with respect to the prevention of co-intercalation, the structural stability, and the surficial passivation composition (SEI).

The Supplementary Figure 28c and d verify the outstanding performance of the graphite half cell in the DPE electrolyte. It shows a 93.91 % capacity retention after 1000 cycles and an average CE of 99.90 % starting from the 30th cycle. The voltage profile exhibits minimum graphite electrode slippage and

overpotential increase². The typical Li⁺ intercalation stages are also well observed. These results demonstrate the great long-term structural and interfacial stability of the graphite anode.

A comparative study was then performed with the DPE, the regular DME ether electrolyte and the commercial carbonate electrolyte (1 M LiPF₆ in EC/EMC (3:7; volumetric)). In Supplementary Figure 29a, b and c, the DPE and carbonate electrolytes shows typical Li⁺ intercalation profiles into the graphite, while the DPE exhibits less overpotential. The DME electrolyte, however, shows a different co-intercalation behavior with the reactions occur within a wide potential window³. The graphite anodes were characterized by the XRD after 10 slow CV cycles (Supplementary Figure 29d). Its layered structure was well preserved from the DPE and the carbonate electrolyte, while the DME electrolyte destroyed it and left a broad (002) peak with a largely diminished intensity. It also causes severe electrode pulverization due to the material deformation. Therefore, the ether co-intercalation can be completely prohibited in the DPE electrolyte, which is demonstrated beneficial for preserving the graphite layered structure and minimizing the capacity/active material loss during the long cycle.

On the other hand, the SEI is critical to protecting the graphite anode from the parasitic reactions and prolonging its cycle life. The depth-dependent XPS were used to analyze the SEI chemical compositions. The elemental concentration profiles in the Supplementary Figure 30a, b and c show stark differences between the carbonate-LiPF₆ and the ether-LiFSI electrolytes, where the former contains more C and O species while the ether systems show more F and S products via salt decomposition. The SEI from the carbonate electrolyte is mostly consist of organic components with C-O and C=O groups. The DPE and DME show similar inorganic compositions, but the DPE still facilitates the anion decomposition such as generating more S-O components in the O 1s signal and preventing the formation of S_n²⁻ species in the S 2p result. Like the LMA, such inorganic SEI derived by the anion decomposition can largely protect the graphite surface and suppress the side reactions. It is regarded as the main contribution to the observed high and stable CE performance.

Overall, we can conclude that tailoring the solvation structure, especially reducing the solvating power of ether solvent can dramatically improve the performance of graphite anode and thus the LIB within ether electrolyte. The direct impact of solvation structure on the inhibition of solvent co-intercalation protects the anode from the structural collapse and the active material loss. The selectively generated robust inorganic-rich SEI layer realizes a well-stabilized interphase and prohibits the increase in the overpotential and interfacial impedance.

We hope the discussions well interpret the impact of ether electrolyte solvation structure on the graphite performance. But due to the limited space in the main text, we decided to put this section into the SI (Page 22-24).

Supplementary Figure 28. Long cycle stability of graphite-NCM811 full cell and graphite half cell with 1.8 M DPE electrolyte. The capacity and CE of the full cell (a) and graphite half cell (c) over 1000 cycles. **b** and **d**, the corresponding voltage profiles.

Supplementary Figure 29. The impact of solvent solvation strength on the graphite anode in terms of the electrochemical performance and the structural evolution.

Supplementary Figure 30. The detailed XPS fine spectra analyses of the graphite SEI formed in different electrolytes.

Comment 3. Supplementary information – Materials (Lines 5-6): More details on Positive electrode processing and parameters (porosity/tortuosity, calendaring) could be provided. (see refs. 3,4). A discussion on electrolyte invasion in the electrode depending on surface tension/viscosity and electrodes exposed surface could be added in the main text.

Response: We thank for this comment and the recommended helpful references. Here, we provide the detailed processing parameters of the cathodes and anode used in this study:

For the NMC811 cathode, the as-coated density/porosity was 2.40 - 2.42 g/mL, porosity of 45.0 % – 45.5%. It was then calendered to a thickness of 127 - 128 μm , corresponding to a density of 3.04 - 3.15 g/mL (variation due to tolerances in mass loading), and a porosity of 28.4 % - 31.0 %. It was densified by passing through an 80 °C calender.

The graphite anode had an as-coated porosity/density of 51 %/1.05 g/mL. It was then calendered to a density of 1.42 - 1.47 g/mL, and a calculated porosity of 32 % - 33 %. The anode was also densified via calendering at 80 °C.

For the NCM811 cathode provided by the Argonne National Laboratory, it has a porosity of 37.3 %, calendered thickness of 52 μm and area loading of 8.21 mg/cm^2 .

As discussed above, the nonpolar ether electrolytes have lower surface tension and viscosity compared to the DME and DIG. It is beneficial for the electrolyte wetting the pore surface of the separator and electrodes as well as infilling into their tortuous microstructures. Therefore, it contributes to the improved long cycle performance of the LMB full cells, especially under the lean electrolyte condition (3 g/Ah) that requires the complete wetting of electrodes and separator with very restricted amount of electrolyte.

The electrode parameters have been integrated into the Methods section. We also added the discussion of electrolyte invasion into the electrodes to the main text on page 13 line 5. The suggested papers are also cited.

Comment 4. Do the authors apply any surface treatment/washing to the Cu foil before Li-metal is electrodeposited and / or to electrodeposited Li?

Response: Thanks for this comment. In this study, the Cu foil was subjected to no special treatment. The Cu foil purchased from MTI corp. was sonicated for 10 min with water and ethanol and then vacuum dried at 80 °C before use.

The Cu foil treatment information has been integrated into the Methods section.

Comment 5. Which are the properties (Porosity, tortuosity, surface coating) of the separators used? Do the authors believe that the choice of solvent might have an effect on the (local) microviscosity and thus on the Li-ion transport properties / solvation of the electrolyte? (see refs. 1,5,6)

Response: We thank the review for this comment and the useful reference. The separator used in this study was Celgard 2500 (0.55 porosity, 2.5 ± 0.2 tortuosity⁴) without any surface coating. Indeed, we agree that the local microviscosity is supposed to be an important factor affecting the Li^+ transport through the electrolyte. As suggested by the results of the work from Saito et al⁵, the microviscosity is considered as the combination of the (1). van der Waals interactions of the solvate species with the environment, (2). the cation-anion coulombic interactions, and (3). the interactions between the cation and attractive sites on the separator. The weakly solvated electrolyte enables the strengthened (2) and (3), which contributes to the reduced ion diffusivity. In our case, the ether electrolytes are also inevitably subjected to such influences. Especially, the weak coordination and ion-aggregating behavior in the DPE and DEE electrolyte slow the Li^+ transport due to the abovementioned factor (2) and (3). It is also reflected as the lower ionic conductivity and the larger battery overpotential according to our newly added rate study (Supplementary Figure 5 and 27).

In the meantime, we think our current data cannot confirm the coulombic interactions between the cation/anion and separator can directly alter the solvation structure. But we believe that it would be a very meaningful topic for an individual study in the future, as most of the electrolyte solvation behavior studies are carried out without considering the influence of the separator membrane.

We have added the discussion in the revised main text (Page 6, line 16)

Supplementary Figure 5. The rate performance study of the NCM811 half cells with the different ether electrolytes. **a**, The capacity of each half cells at different cycling rate. **b-e**, the corresponding voltage profiles.

Supplementary Figure 27b-e. The voltage profiles of the full cells with the studied electrolytes under increasing C rate.

Comment 6. Page 15, line 27 (supplementary Figure 21): can the authors provide (at least an approximate) values of contact angle – I suppose the pics were taken in glovebox atmosphere. The conditions (Temperature, pressure, atmosphere) could be specified in caption.

Response: We thank the reviewer for this comment. We tried our best to collect the electrolyte contact angle (CA) on the Li surface during the revision. Since we don't have a dry room, the experiment should be performed in our Ar-filled glove box. We thus could not use the professional instrument to collect and analyze the images. Instead, the electrolytes were dropped on to the Li metal surface in the glove box (25 °C) and the pictures were taken through its glass shield. The DPE and DEE show a complete wetting on Li, which makes it infeasible to determine the CA value. The DME and DIG have much larger and measurable CAs (appropriately 36° and 48°, respectively). The images were processed by the ImageJ software with the contact angle plugin.

Supplementary Figure 25. Wettability testing of ether electrolytes on the LMA surface. **a-d**, The top view images of the electrolytes on the Li surface. **e-h**, The side view images and the approximate contact angle. The testing was performed in the Ar-filled glove box (25 °C), and the pictures were taken through its glass shield. 15 μL different electrolytes were dropped on to the Li metal surface. The DPE and DEE show a complete wetting on the Li, which makes it infeasible to determine the CA value. The DME and DIG images were processed by the ImageJ software with the contact angle plugin.

We have added the results in the SI page 21

Comment 7. The experimental section (Supplementary Information page 1, lines 29-30) describing the Raman spectroscopy analysis in Figure 2e (main text) should include the objective magnification, spectral resolution and the grating used for the spectrographic analysis. Do the authors perform the analysis at different pump power to exclude any thermal effects?

Recent literature on operando Raman analysis of battery electrolytes suggested that changes in the electrolyte solvation structure are correlated to the SEI formation. (refs. 7 and 8) These reports could support the claims of the authors and should be cited.

Response: Many thanks for this suggestion and the helpful references. The Raman microscope has the 50 times objective magnification and the 600 lines/mm grating. The spectral resolution of the collected Raman data is 1 cm^{-1} . We performed the Raman analyses at different laser power during the paper

revision to exclude the thermal effects. As shown in Supplementary Figure 34, the four ether electrolytes with 1.8 M salt concentration were subjected to the Raman analyses with increasing laser power from 0.5 to 7.5 mW. Increasing the laser power generates higher signal intensity but induces no peak position shift. Therefore, there is no observable thermal effect from the Raman laser under the testing conditions in this study. The Raman analyses which were generated with the 7.5 mW laser in this study are supposed to be reliable.

Supplementary Figure 34. The laser power dependent study of the Raman spectra. The four ether electrolytes with 1.8 M salt concentration were subjected to the Raman analyses with increasing laser power from 0.5 to 7.5 mW. The exposure duration and scan times were kept consistent at 7.5 s and 50 times.

We have integrated these information, data, and discussions into the revised manuscript (Page 15, line 5) and SI (Page 27). The suggested references have been cited.

Minor:

- Figure 1c , inset: hard to read.
- Page 3, line 20: coma missing between 8 and 10 (8,10)
- Page 5, line 4: a citation to support the steric hindrance argument would be beneficial.

Response: Thanks for these suggestions. We have improved the qualities of all the figures including their font size and picture resolution.

A comma has been added between reference 8 and 10 at the designated place.

We also added a citation regarding the steric hinderance (**Chen, Y. *et al.* Steric Effect Tuned Ion Solvation Enabling Stable Cycling of High-Voltage Lithium Metal Battery. *J. Am. Chem. Soc.* 143, 18703–18713 (2021))⁶.**

Reviewer #2

The manuscript by Li *et al.* reports on a novel approach of using highly nonpolar electrolytes for Li/NMC batteries. The data presented demonstrates higher coulombic efficiency for cells with more nonpolar ethers as compared to the more polar ethers used in this study. Electrolytes solvation structure in the bulk and at the interface are explored by several experimental and theoretical methods. Authors also report the effect of electrolyte structure on the electrode/electrolyte interface composition. This work explores the stabilization of metal lithium anode and high voltage cathodes, which is a very relevant topic. However, the main concern is that electrolytes employing nonpolar solvents have much lower conductivity, that would affect battery performance, which is not discussed in detail in the manuscript.

The following concerns and questions should be addressed by authors before publication is possible:

Comment 1. Figure 1b shows electrochemical stability windows of the electrolytes, however DPE and DEE based electrolytes have about 10 times lower conductivity (as reported in supl. Fig. 3). Was ohmic drop IR correction performed in this measurement? Without ohmic drop correction the stability windows estimation is not reliable. Please include IR drop correction in the experimental details and the LSV curves with and without IR correction.

Response: We appreciate this scientific comment from the reviewer. Here, we have performed the IR compensation LSV on the studied ether electrolytes with the Gamry Potentiostat on its Current-Interrupt mode. The results are compared with the ones without correction in the Figure 1c and Supplementary Figure 3. Overall, the IR-corrected LSV profiles shift toward lower potential. They indicate that the determined electrolyte oxidation potential is supposed to be lower when considering the electrolyte bulk resistance. However, we noticed that the corrected DPE and DEE results are not largely deviated from the uncorrected results. We think the reason is that both studied nonpolar electrolytes induce very low anodic current densities (less than 2 $\mu\text{A}/\text{cm}^2$) throughout the testing up to the cut off potential (6 V). Considering the relationship between the voltage deviation (ΔV), the bulk electrolyte resistance (R_u) and the anodic current density (I_{cell}) of the cell ⁷:

$$\Delta V = V_{\text{measured}} - V_{\text{corrected}} = I_{\text{cell}} \times R_u$$

the ΔV becomes negligible despite the large cell resistance when the reaction anodic current density is very small.

We hope that the results with IR correction could now provide reliable conclusions regarding the oxidation stability of the ether electrolytes on the Al working electrode. The discussion has been included in the SI page 4.

Figure 1c. LSV of the four electrolytes with Al working electrode with and without IR correction. The scan rate is 0.5 mV/s

Supplementary Figure 3. The zoomed-in plot of LSV analysis of four electrolytes on Al electrode.

Comment 2. The highest C-rate used in the study is C/3, which is a relatively low C-rate for commercial applications. Did authors try to perform higher C-rates measurements, for example 1C? Is C/3 the maximum C-rate possible due to the low electrolyte conductivity? Please add the discussion about performance limitations due to the low electrolyte conductivity.

Response. Thanks for this great suggestion. We have performed the rate studies on both NCM811 half cells and full cells with the ether electrolytes.

The maximum C rate of the nonpolar electrolytes in this study is certainly not C/3. As shown in Figure 1f and Supplementary Figure 5, stable cycling of the half cells at 1 C rate or higher can be realized. The DPE electrolyte still demonstrate the highest cycling stability at 1C thanks to the easier desolvation and optimized interfacial properties (interphase composition and EDL structure), while the DME and DIG shows fast capacity decrease at 1 C and the sign of early battery failure at 2 C rate. However, due to the strong ion aggregating behavior and the resulted small ionic conductivity, the DPE exhibits consistently higher overpotential and slightly less capacity at each studied C-rate.

In terms of the full cells with controlled amount of electrolyte and Li anode (Supplementary Figure 27), the DPE and DEE electrolyte still delivered stable cycling at the 1 C rate. The DME and DIG cells,

instead, failed to complete the rate study because their inferior plating/stripping efficiency caused quick Li anode consumption at high current densities. Using thick Li anode for them managed to obtain their rate performance. However, they still show faster capacity decrease and battery failure despite the excessive amount of Li. On the other hand, it is also important to note the enlarged ionic transport limitation in the full cell configuration with heavy loading. The DPE requires large overpotential at 1C, where around 40 % of capacity were charged during the constant-voltage step. In contrast, the DME electrolyte with superior ionic conductivity demonstrates the lowest overpotential. Therefore, our results verify that the rate performance is governed by both electrolyte ionic conductivity and the interfacial behaviors. Both factors are supposed to be optimized for the stable high-rate battery performance. The DPE or DEE electrolytes which feature fast interfacial kinetic and high CE also inevitably suffer from their limited bulk ionic transport due to the strong Li⁺-anion interactions. They will meet their limitations when increasing the rate and electrode loading to an even higher level.

We have integrated these discussions into the main text (Page 5, line 23; Page 6, line 16; Page 12; line 25) to highlight the important relationship between the interfacial behavior and the ionic conductivity when tuning the electrolyte solvation structure.

Figure 1f. The NCM811 half cells cycled at 1 C rate with studied electrolytes.

Supplementary Figure 5. The rate performance study of the NCM811 half cells with the different ether electrolytes. **a**, The capacity of each half cells at different cycling rate. **b-e**, the corresponding voltage profiles.

Figure 6b, Rate study of the full cells with different electrolytes.

Supplementary Figure 27. The rate study of the practical LMB full cells with the different electrolytes. **a**, the quick failure of the DME and DIG full cells with limited Li anode during the rate study. **b-e**, the voltage profiles of the full cells under increasing C rate. The DME and DIG used thick Li anode.

Comment 3. Please add the Raman spectra resolution in the experimental details.

Response: Many thanks for this comment. The Raman spectral resolution was also mentioned by the first reviewer. It is 1 cm^{-1} in this study. We have updated the Raman experimental details in the manuscript as:

Raman spectroscopy studies were performed with Thermo Fisher Scientific DXR Raman microscope with a 633 nm laser source. The Raman microscope has the objective magnification of 50 \times and the grating of 600 lines/mm. The spectral resolution of the collected Raman data is 1 cm^{-1} . 7.5 mW laser power, 7.5 s exposure duration, and 50 scans were used for Raman spectra acquisition.

Comment 4. Ex situ XPS measurements provide conclusions that less polar electrolyte leads to more inorganic SEI. Please add in the experimental details the procedure for ex situ samples preparation. Were

the electrodes washed with the solvent, and in that case with which solvent? Is there any possibility that detected F-rich layer is due to the electrolyte salt residue rather than SEI? If the electrodes were washed with non-polar solvent, it is more probable that there would be salt residues, that could form LiF as a result of argon etching.

Response: We thank the reviewer for bringing up this good point. We are also aware of the recent publications regarding the salt residue and its transformation to LiF under Ar plasma etching⁸.

The cycled cathode and anode were rinsed in 1 mL of the corresponding fresh ether solvent for 10 min and 3 times during the sample preparation for the XPS.

In terms of the SEI composition from the XPS, we would like to note that the surficial XPS results before the Ar etching also support the LiF-rich or F-rich SEI/CEI from the nonpolar electrolyte. More importantly, the nonpolar ethers (DPE and DEE) used in this study can dissolve up to ~ 4 M LiFSI (salt to solvent molar ratio 1:2), as demonstrated in Figure 2e. They are thus supposed to adequately wash out the residue salt, given 3 times of washing with 1 mL of fresh solvent per each 12 mm electrode disk.

To confirm our hypothesis, we used ¹⁹F NMR to quantitatively track the residue salt concentration from the solvent samples after the serial washing of a completely electrolyte-wetted cathode. A direct comparison was made between the DME and DPE solvent (Supplementary Figure 35). As shown in the Supplementary Figure 35d, DME and DPE exhibit similar ability of washing out the residue salt. The concentrations from the sample at the third time of washing dropped down below 1 % (0.27 % DME vs. 0.52 % DPE) of the original value, which we believe, can be regarded as the removal of most residue salt.

Supplementary Figure 35. The washing process of the cycled electrodes and using NMR to confirm the electrode can be sufficiently rinsed by the nonpolar ether. **a**, the schematic of the applied washing process in this study to prepare the XPS samples. **b-c**, The ¹⁹F NMR of the serial washing solvent, where

the peak area is the direct indication of the concentration. **d**, The normalized salt concentration of the washing solvent.

We hope this additional NMR study addresses the reviewer's concerns about the residue salt in the electrode and its impact on the XPS results. We have integrated this part in the revised main text (Page 15, line 25) and SI (Page 28).

Comment 5. Could SEI components low solubility in the electrolyte be a contributing factor for the SEI stability? Could this also explain the results of electrolyte exchange studies, considering that SEI components are likely to be more soluble in DME as compared to DPE?

Response: We appreciate the reviewer for this great comment. We also realized that there are recent publications highlighting the importance of SEI/CEI solubility⁹. During the paper revision, we utilized the method reported by Jin et al.⁹ to assess the solubility of the CEI from DPE electrolyte into the pristine DPE and DME solvent.

As illustrated by the Supplementary Figure 15a, one single NCM811 cathode was cycled in the DPE electrolyte at 0.1 C for 10 times and then washed by the DPE solvent and dried. It was next cut into 3 identical pieces. One piece was subjected to soaking in 2 mL DME solvent for 40 h, another one was soaked in 2 mL DPE solvent for 40 h, and the third one was not treated. The 3 samples were then analyzed by XPS.

In the Supplementary Figure 15b-e, the chemical compositions and their relative ratios from the C 1s, O 1s, and S 2p signals show minimum differences between the samples before and after soaking in either DPE or DME solvent. Interestingly, only LiF, which is indicated by the F 1s signal at ~ 685.0 eV, shows decreased concentration after soaking in the solvents. The DME dissolves more of this individual component than the DPE. In this case, we can cautiously draw several conclusions with this intriguing result. First, the LiFSI salt decomposition products such as the SO_x and SO_xF_y (indicated by the O 1s, S 2p and the 687.5 peak of F 1s) possess very low solubility in both polar and nonpolar electrolyte/solvent, even lower than the well-known “non-soluble” LiF. The DPE electrolyte which induces the enrichment of these components in the SEI and CEI is very advantageous for the stable long cycle performance. Second, the LiF component is qualitatively demonstrated with higher solubility in the DME electrolyte/solvent than DPE, which might be partially attributed to the CE drop during the electrolyte exchange study. However, we believe it is important to note that the decreasing of the LiF peak (LiF dissolution) is mainly due to the very thorough soaking process. A large amount (2 mL) of ether solvents were used to soak a tiny piece of cycled cathode (~ 8 mg NCM811), and the CEI is an extremely small fraction (nanometer level thick) of the entire electrode. Thus the decreasing of the LiF peak becomes so obvious, despite the well-known low solubility of LiF in ether (e.g. 0.09 mM in tetrahydrofuran¹⁰). In this case, we believe the SEI/CEI formed in the DPE electrolyte can be sufficiently regarded as low solubility, considering the most soluble specie turns out to be the LiF. Therefore, we would still consider the EDL change as the main reason of the CE shifting during the electrolyte exchange study.

Besides, the C-F peak in the F 1s and C 1s signal should originate from the non-soluble FSI⁻ decomposition products, instead of the salt residue, as their concentration persist after the thorough washing and soaking processes. We think this can correlate with our respond to the previous comment.

We have added the abovementioned discussions in the revised SI (Page 13) and main text (Page 8, line 30).

Supplementary Figure 15. Studying the solubility of the CEI which was formed in the DPE electrolyte in the DPE and DME solvent. **a**, the schematic illustrating the experiments. **b-e**, the C 1s, O 1s, F 1s, and S 2p XPS fine spectra from the studied samples.

Comment 6. The interfacial MD simulations on cathode surface were performed using graphene slab as electrode interface. Could you please comment how comparable and relevant the charged graphene interphase to the NMC cathode, could the nature of interface influence the results?

Response: We thank the reviewer for this important question. To the best of our knowledge, modeling work in the literature involving NMC cathode has been mainly conducted using *ab initio* molecular dynamics (AIMD) simulations. However, AIMD simulations require substantial computational resources, which limits the size of the system that can be modeled. In our work, we are interested in modeling the

number density of the different components of the electrolyte and in understanding how the solvation structure of the cation changes near the surface in the presence of an applied voltage compared to the bulk. Therefore, obtaining a statistical average of the solvation structure for a complex and multi-component system like our electrode/electrolyte system is not feasible at present using AIMD simulations. Even for the published AIMD simulation work, Li_xNiO_2 has been used as an approximation to the NMC cathode to simplify modeling considerations (e.g. sampling of Mn and Co sites)¹¹.

We also respectfully note the interfacial MD simulations in this work are meant to provide a qualitative comparison of the behavior near the interface in the DPE system compared to that in the DME system. Therefore, while the exact number densities might change if we use NMC as the model cathode, we believe the qualitative analysis and conclusions made from the interfacial MD simulations will remain the same. We also note that force field parameters required to model the NMC cathode using MD simulations are not available in the literature, which makes it infeasible to model this cathode material. Recent work using NMC cathode for a similar electrolyte system composed of LiFSI/LiNO₃ in DME has used graphene as a model cathode in classical MD simulations. The authors in this work reported that the interfacial modeling results using graphene were consistent with their experimental XPS results. In addition, the authors ran additional AIMD simulations on a simplified system that contains only the solvent and found that the solvent number density from AIMD simulations is very close to that obtained from MD simulations¹².

We have added this discussion as a supplementary note in the SI page 29.

References

1. Sun, Y., Radke, C. J., McCloskey, B. D. & Prausnitz, J. M. Wetting behavior of four polar organic solvents containing one of three lithium salts on a lithium-ion-battery separator. *J. Colloid Interface Sci.* **529**, 582–587 (2018).
2. Dose, W. M., Xu, C., Grey, C. P. & De Volder, M. F. L. Effect of Anode Slippage on Cathode Cutoff Potential and Degradation Mechanisms in Ni-Rich Li-Ion Batteries. *Cell Reports Phys. Sci.* **1**, 100253 (2020).
3. Yao, Y. X. *et al.* Regulating Interfacial Chemistry in Lithium-Ion Batteries by a Weakly Solvating Electrolyte. *Angew. Chemie - Int. Ed.* **60**, 4090–4097 (2021).
4. Landesfeind, J., Hattendorff, J., Ehrl, A., Wall, W. A. & Gasteiger, H. A. Tortuosity Determination of Battery Electrodes and Separators by Impedance Spectroscopy. *J. Electrochem. Soc.* **163**, 1373–1387 (2016).
5. Saito, Y., Takeda, S., Morimura, W., Kuratani, R. & Nishikawa, S. A Selective Interaction between Cation and Separator Membrane in Lithium Secondary Batteries. *J. Phys. Chem. C* **121**, 23926–23930 (2017).
6. Chen, Y. *et al.* Steric Effect Tuned Ion Solvation Enabling Stable Cycling of High-Voltage Lithium Metal Battery. *J. Am. Chem. Soc.* **143**, 18703–18713 (2021).
7. Gamry Instruments. Understanding iR Compensation.
8. Yu, W., Yu, Z., Cui, Y. & Bao, Z. Degradation and Speciation of Li Salts during XPS Analysis for Battery Research. *ACS Energy Lett.* **26**, 3270–3275 (2022).
9. Jin, Y. *et al.* Low-solvation electrolytes for high-voltage sodium-ion batteries. *Nat. Energy* **7**, (2022).
10. Wynn, D. A., Roth, M. M. & Pollard, B. D. The solubility of alkali-metal fluorides in non-aqueous solvents with and without crown ethers, as determined by flame emission spectrometry. *Talanta* **31**, 1036–1040 (1984).
11. von Aspern, N. *et al.* Methyl-group functionalization of pyrazole-based additives for advanced lithium ion battery electrolytes. *J. Power Sources* **461**, 228159 (2020).
12. Zhang, W. *et al.* Engineering a passivating electric double layer for high performance lithium metal batteries. *Nat. Commun.* **13**, (2022).

REVIEWERS' COMMENTS

Reviewer #1 (Remarks to the Author):

The Authors successfully improved a high quality piece of research and my points were all completely addressed. I would recommend this article to be published in Nature Communications.

minor:

line 16 page 10 revised manuscript: maybe the authors mean EDL instead of ELD.

Reviewer #2 (Remarks to the Author):

The authors have very carefully revised the manuscript and have incorporated suggestions provided by reviewers. The discussion in the rebuttal letter is sound and very detailed. Authors have also performed several additional measurements and analysis (surface tension and viscosity, IR drop compensation, NMR analysis of electrode washing procedure). These additional measurements and detailed discussion have significantly improved the manuscript, as well as strengthened the discussion and conclusions.

I recommend the revised manuscript to be published as it is.